# Trustworthy AI for medical decisions: Adversarially robust and fair machine learning prediction for Parkinson's disease

Junaid Muhammad[1], Mitra Ghergherehchi[1]*, Shiraz Ali[1],
Ho Seung Song[2]*, Nasir Rahim[3]

1 Department of Electrical and Computer Engineering, Sungkyunkwan University, Suwon, South Korea,
2 Catholic Kwandong University, Department of Electronic Engineering, Gangwon-do, South Korea,
3 School of Computing, Gachon University, Seongnam, Republic of Korea

* mitragh@skku.edu (MG); hssong@cku.ac.kr (HSS)

## Abstract

Parkinson's disease (PD) is a neurodegenerative disorder characterized by motor and non-motor symptoms, including tremor, rigidity, and postural instability. Machine learning (ML) models have shown promise for the diagnosis of PD; however, many existing approaches do not explicitly address fairness and robustness. As a result, these models can lead to biased outcomes across demographic groups and vulnerability to adversarial attacks. In this study, we used the Parkinson's Progression Markers Initiative (PPMI) cohort, which includes clinical and demographic information from 1,084 participants spanning diverse age, sex, and racial groups. Our study addresses the key challenge of developing robust and equitable ML models to diagnose the progression of PD. We evaluated the performance of two fairness-optimized classifiers, namely, Random Forest (RF) and Decision Tree (DT). To evaluate model vulnerability, we applied adversarial techniques, specifically label leakage and data poisoning attacks, which simulate intentional or erroneous data alterations that can amplify biases and degrade accuracy. These adversarial manipulations substantially degraded model performance; specifically, DT accuracy declined by more than 10% between sensitive groups. The accuracy of the RF model decreased by 20%. Moreover, under attack, fairness metrics such as Statistical Parity Difference (SPD), which looks at differences in the chances of getting a positive prediction across demographic groups, and Equal Opportunity Difference (EOD) for differences in true positive rates between groups, both showed a decline. This pattern suggests that adversarial perturbations increased bias and widened performance disparities across demographic groups. Our results demonstrated that adversarial attacks increased the incidence of false positives and false negatives, thereby lowering the accuracy and fairness of the PD diagnostic predictions. These findings underscore the urgent need for robust and fairness-aware defenses in medical AI to mitigate

**Data availability statement:** The dataset used in this study is sourced from the Parkinson's Progression Markers Initiative (PPMI), a publicly accessible, longitudinal research study sponsored by The Michael J. Fox Foundation. Access to the dataset can be requested via the official PPMI portal at: http://www.ppmi-info.org/data. All data used in our analysis were obtained in compliance with PPMI's data use agreement. All variables/features used in our analysis are documented in our public repository (https://zenodo.org/records/18489583).

**Funding:** This research was supported by the Regional Innovation System & Education (RISE) program through the Gangwon RISE Center, funded by the Ministry of Education (MOE) and Gangwon State (G.S.), Republic of Korea (Grant No. 2025-RISE-10-001 to H.S.S.).

**Competing interests:** The authors have declared that no competing interests exist.

racial, age, and gender disparities and ensure a reliable clinical decision-making process.

## 1 Introduction

Neurodegenerative disorders, such as Parkinson's disease (PD), are conditions that people acquire as part of their lifelong journey. It often attacks middle-aged to older individuals. Common symptoms include tremors [1], bradykinesia (slow movements) [2], rigidity, postural instability, and gait difficulties [3] resulting from loss of instinct, movements, speech, and handwriting problems [4]. Other symptoms include hallucinations, depression, insomnia, and low blood pressure [5]. There is no cure for PD; however, surgery and medical options can manage symptoms. However, like any other drug, all drug options produce side effects that are undesirable in a normal daily life scenario [6]. Health care has been discriminated against and treated unfairly throughout history, and the development of Artificial Intelligence (AI) will only make it more able to be unfair [7].

Modern AI-driven systems can identify statistically significant patterns in data that humans cannot discover [8]. This accounts for more accurate data analysis and modeling. Although these systems are subject to frequent errors, their modeled outcomes outperform established approaches. It will lead to presenting a real challenge to data and domain experts [9]. The emerging problem from data-intensive methods is the limited ability to discern biases in ML projections. Medical diagnosis [10], college admissions [11], loan distribution, prediction of recidivism, recruitment, online advertising, facial recognition, language translation, recommendation engines, fraud detection, credit decisions, pricing, and fake news detection [12]. At the same time, research has not yet examined the relationships of race, age, and gender [13] with the diet of PD in cognitive capabilities [14].

Initially, men exhibit more stiffness and show more sleep disturbances during rapid eye movement compared to women, while women suffer more from dyskinesia and depression [15]. Non-motor symptoms and quality of life related to the health of PD patients differ by sex [16]. From the onset of PD, the quality of life related to women's health is most adversely affected by fatigue and depression. There is a theoretical explanation for the variations in quality of life experienced by people with PD [17]. To our knowledge, no research has examined the potential bias in identifying vulnerable groups of patients using PD-based ML, particularly those designated by age, gender, or ethnicity, who may be disproportionately targeted by such prejudice. [18].In the context of PD, where the accuracy of diagnosis significantly impacts patient outcomes, this is especially crucial. The resilience of AI models, as emphasized in [19], guarantees uniform performance even when dealing with varied and volatile clinical datasets. The present study highlights the simultaneous importance of equity and resilience in the development of reliable AI systems for healthcare. Through the integration of robustness into our fairness-aware methodology, our objective is to make a valuable contribution towards the development of AI systems that possess not only impartiality but also resilience and dependability in operational

medical environments. The ML process is characterized by two main sources of inadvertent bias [20,21]. Models can learn from incorrect or biased data, which means that they will find relations but not real-world patterns and produce results with varying reliabilities [22]. Even though today's data accurately reflects patterns, it may still contain biases, making AI susceptible to legal challenges for unfairness [23].

Our study thus investigates the impact of data bias on both diagnostics and treatment regimes, using ML models to ensure consistent performance for a wide range of clinical scenarios, and provides solutions to address these biases. This research's primary focus is the development and testing of a robust and equitable ML system for identifying PD. In contrast to previous work, we thoroughly test our model's performance in hostile scenarios, such as label-leak attacks and poisoning. We also use preprocessing and fairness measures to account for biases related to age, gender, and race. A novel and beneficial step toward applying AI to make trustworthy medical decisions is the two-pronged perspective of strength and fairness.

1. The primary contribution of this paper is the early study to jointly examine fairness-aware preprocessing and adversarial robustness in ML models for Parkinson's disease, providing a framework that reduces demographic bias while testing vulnerability to data poisoning and label-leak attacks.

The remainder of the paper is organized as follows. Sect 2 reviews related work on fairness and robustness in medical AI. Sect 3 describes the dataset, evaluation metrics, and the methodological framework. Sect 4 presents the proposed models and adversarial settings. Sect 5 reports the experimental results. Sect 6 Discussion of the research findings, while Sect 7 discusses the limitation of the proposed research. Finally, in Sect 8 Conclusion and future work are reported.

## 2 Related work

This section provides a comprehensive overview of the literature on fairness, examined through multiple perspectives. Initially, we discussed biases and the critical importance of trustworthiness in the medical field. Subsequently, we shift focus to research on fairness. In addition, we discuss recent developments in clinical decision support systems (CDSS) and fairness in the context of ML and DL [24]. According to research on adversarial resilience in ML models, adversarial attacks can worsen or disclose biases in medical data [25]. The article discusses strategies for protecting ML models from attacks while ensuring fairness.

### 2.1 Profound bias patterns

Prior to explicitly discussing what fairness is, we can recognize common biases that lead to unfair conduct in commercial ML systems; a few examples of biases in ML are discussed as follows. Training data often inherently reflects existing human biases. In situations where bail and parole decisions are involved, predictions might help tackle problems like recidivism [10]. Predicting whether an ex-offender would commit a new crime within a given time frame is their goal. But instead of using convictions, they use arrest records. Those involving drug offenses are disproportionately associated with minority groups, which are often the targets of increased enforcement [26]. ML algorithms are made to match the data by automatically reproducing the bias that already exists in the data that leads to propagation rather than elimination of existing prejudices [27]. It is important to minimize the average error, which benefits the majority group as their data is more consistent and predictive of the results, which reduces the general error. This leads to disproportionate errors affecting minority groups within diverse populations, as limited access to resources can compromise the reliability of their data [28]. Addressing these disparities through further research is both urgent and essential.

The inputs to the prediction algorithm are influenced by its past behaviors in critical areas like medication trials, etc. In a wide range of professions, including law enforcement and healthcare, it is common practice to make decisions only after gathering insufficient information [29,30]. Exploratory behavior, which involves performing actions that may not always be

optimal to acquire additional data, is the optimal approach to maximizing information acquisition in such circumstances, according to learning theory. First, we must determine whether the expenses of exploratory activity disproportionately harm any population. Second, inadequate measurements for an individual patient can be unethical in medical studies, limiting learning and perpetuating unfairness.

## 2.2 Trustworthy AI in medicine

One of the most significant barriers to the widespread adoption of AI is the public's skepticism toward ML. To earn trust, data scientists must go beyond traditional performance metrics such as prediction accuracy and prioritize ethical considerations. Minimizing bias and promoting fairness while maintaining robustness and transparency are essential to foster a future in which ML algorithms serve as reliable collaborators for humans [31,32]. Building confidence in AI systems requires attention to critical factors throughout the entire lifecycle of a predictive model, including data collection, preprocessing, feature engineering, model training, and optimization. Equally important are the stages of testing, deployment, and ongoing monitoring. Together, these interconnected steps form what is known as the "chain of trust" [33,34].

## 2.3 Fairness in ML-based approaches

The term "fairness" encompasses a wide range of scenarios, from simple binary decisions to complex real-time policy control situations. It is a complex and context-dependent concept dictated by local societal norms. Ensuring fairness within an organizational context is challenging. Notably, fairness can be mathematically expressed in 21 ways [35]. Conflicts in how fairness is measured can highlight different aspects of fairness, but they also have implications for test results. Recent work on fairness has shown that satisfying multiple reasonable fairness axioms is impossible. A decision tree to determine whether an organization can achieve fairness, developed by the University of Chicago, could be useful [36].

## 2.4 ML and DL approaches for clinical decision support system

Medical AI research has increasingly focused on applications in real-time clinical settings. In this context, researchers have developed accurate PD's clinical decision support systems (PD-CDSS) by leveraging ML and DL techniques, as illustrated in Fig 1. However, such models often underperform in real-world clinical practice compared to initial expectations. Recent efforts have therefore emphasized the development of high-quality medical CDSS models that require less patient data while integrating a wider range of ML algorithms. Although a COVID-19 diagnostic technique was proposed by [37] as a consistent approach, it diverged from established methodological criteria and definitions. Alongside renewed interest in neural network security, researchers have also developed interpretable DL models that address domain-specific features and associated security challenges. Nevertheless, given current technological limitations, no definitive solution exists for comprehensive risk assessment in AI-driven clinical applications [38]. Several critical questions remain regarding how end users perceive and adopt these systems, as well as the perceived limitations in their

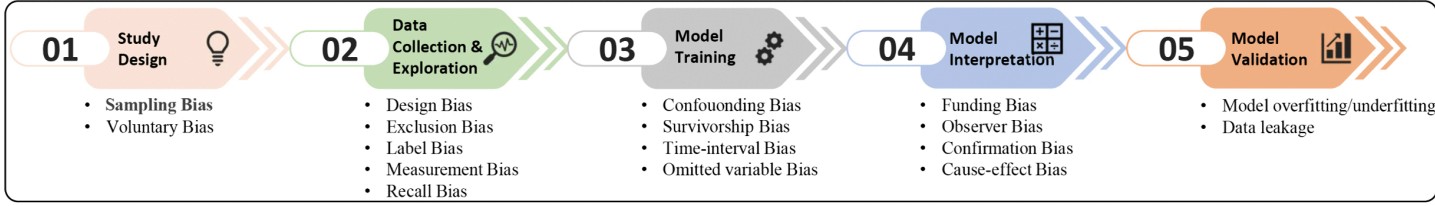

**Fig 1**. The figure illustrates the various steps of a ML pipeline and highlights considerations related to model fairness.

design and deployment. Despite substantial investment in this area, such challenges continue to hinder widespread clinical adoption. The effective integration of AI-enhanced clinical decision support systems (AICDSS) into healthcare therefore requires rigorous empirical investigation [39]. Furthermore, fostering sustained trust is a fundamental requirement, grounded in eight core principles that clinical AICDSS should satisfy: justness, accountability, responsibility, robustness, transparency, replicability, security, and privacy. Empirical studies further indicate that end users report greater confidence in automated systems when key attributes such as explainability, security, privacy, and reliability are present, all of which are considered critical for practical deployment. In addition, ongoing research on fairness in ML extends to domains such as ranking algorithms, recommendation systems, and contextual bandit learning.

### 2.5 Comparison of machine learning approaches

PD is a progressive neurodegenerative disorder characterized by an early and accurate diagnosis that remains a clinical challenge, as subtle prodromal signs often elude standard assessments. In response, researchers in Table 1 have turned to ML and DL techniques leveraging neuroimaging, clinical, and biomarker data to improve diagnostic accuracy and forecast disease progression. Initial efforts employed traditional classifiers such as support vector machines (SVM), random forests (RF), and XGBoost, combined with feature selection, to achieve accuracies of 80–83% [41,43]. Zhang et al. [40] advanced this work by integrating convolutional neural networks (CNNs) with long short-term memory (LSTM) models on the PPMI dataset, reporting an accuracy of 85.3 % and an F1-score of 84.5 %. Kim et al.'s proposed CNN-SVM framework, evaluated on both external cohorts and PPMI data, further improved performance to 87 % accuracy and an AUC of 0.89 [36]. More recently, reinforcement learning approaches have been explored to adaptively refine model parameters [33], while ensemble methods have enhanced sensitivity and stability [35]. Fairness-sensitive architectures and adversarial training exemplified by Malik et al.'s random forest strategy framework have also been proposed to enhance robustness [42]. Despite these promising results, most studies focus narrowly on expected performance metrics. They rarely assess equity across demographic groups or examine model behavior under adversarial conditions. This gap erodes trust and risks perpetuating biases when ML systems enter diverse clinical environments. To bridge this gap, we propose a unified framework that integrates rigorous robustness evaluations, including label-leak and data-poisoning attacks, with fairness-aware training protocols. By simultaneously measuring predictive accuracy and demographic

**Table 1**. Comparison of recent ML approaches for PD detection/progression (2021–2025), including our work.

| S.No | Study (Year) | Method(s) | Dataset | Key Results/Metrics |
|---|---|---|---|---|
| [40] | Zhang et al. (2021) | CNN + LSTM | PPMI | Accuracy: 85.3%, F1-score: 84.5% |
| [36] | Kim et al. (2021) | Hybrid CNN + SVM | PPMI + External Validation | Accuracy: 87%, AUC: 0.89 |
| [41] | Martinez-Eguiluz et al. (2022) | SVM, RF, XGBoost | PPMI | Accuracy: 83%, AUC: 0.87 |
| [32] | Paul et al. (2022) | Random Forest + Feature Selection | PPMI + Local Clinical Dataset | Accuracy: 82.1%, Balanced Accuracy: 80.7% |
| [33] | Alam et al. (2023) | Deep Reinforcement Learning | PPMI + Wearable Data | Accuracy: 86.1%, Sensitivity: 85.7% |
| [34] | Rohit Surya et al. (2022) | Decision Tree, RF | PPMI | Accuracy: 80-85%, Fairness metrics improved |
| [35] | Singh & Jain (2022) | Ensemble RF + CNN | PPMI | Accuracy: 86.7%, Robustness improved |
| [36] | Zhang et al. (2024) | Fairness-aware Neural Networks | PPMI | Accuracy: 82.5%, Equalized Odds improved |
| [42] | Malik et al. (2023) | Adversarial training + RF | PPMI | Accuracy: 83.9%, Robustness against attacks |
| Our Work | **Our Work (2025)** | Fairness-Optimized DT, RF with adversarial attack analysis | PPMI | Accuracy DT: 82%, RF: 85% (drops 10-20% post-attack), EOD and SPD decline under attacks |

parity, our approach constructs a comprehensive "chain of trust" for ML-based PD diagnosis, aligning technical validation with clinical and ethical standards.

## 3 Materials and methods

This section first examines the dual challenges of fairness and robustness in medical AI, with a particular focus on PD's applications. The robustness of the model is then explored from two complementary perspectives. The first concerns adversarial robustness, which considers threat models such as label leakage and data poisoning. The second addresses data bias by analyzing demographic skews in age, gender, and race within the PPMI cohort. Finally, this section introduces the evaluation framework. This includes bias metrics, such as Statistical Parity Difference and Equal Opportunity Difference, as well as group fairness principles, is used to assess model equity across sensitive attributes.

### 3.1 Fairness and robustness in medical AI

Although there has been promise in using ML to diagnose PD, biases are introduced by issues such as limited medical data sets, inadequate validation, and a lack of reliable clinical evaluations. As an outcome, to guarantee that an AI algorithm that uses PD symptoms (or risk factors) as input should retain accuracy and dependability to reduce AI bias. Biases about race and ethnicity have been found in a comprehensive systematic review [44]. The assessment also highlighted prejudices toward minorities and language. The discussion focused on issues related to the assessment of stability and dependability to reduce biases, the summarization of validation techniques, and the analysis of large datasets. In this study, we employ an optimized preprocessing technique to mitigate bias in the PD dataset prior to model training. Preprocessing methods aim to correct data imbalances and reduce the disparate representation of sensitive attributes such as age, gender, and race. This approach was selected for its effectiveness in improving fairness while maintaining model performance, as demonstrated in recent literature. We focus specifically on this method rather than in-processing or post-processing techniques to provide a clear and targeted fairness intervention aligned with our experimental goals [45,46].

### 3.2 Model adversarial robustness

The resilience of ML models, particularly DL architectures, against adversarial attacks has emerged as a critical area of research, especially in the context of medical imaging applications. A study employing the Fast Gradient Sign Method (FGSM) on a Vision Transformer model demonstrated that minor image perturbations can lead to substantial degradation in classification performance, with accuracy decreasing from 90.1% to 27.38% [47]. However, the same study showed that adversarial training can significantly enhance the robustness of the model, increasing the precision of the classification to 96.61%. Recent studies [35,48] have further explored quantum adversarial machine learning (QAML), which integrates quantum computing principles with ML to strengthen model defenses against adversarial attacks. QAML exploits quantum properties such as superposition and entanglement to potentially improve robustness against input perturbations. Early findings [49] suggest that quantum-enhanced defense mechanisms may produce improvements in adversarial resilience. Additional investigations [50] have examined the relationship between adversarial robustness and model complexity in medical image classification tasks, with particular emphasis on vulnerability to DeepFool attacks and Jacobian-based Saliency Map Attacks (JSMA). These attack strategies deliberately manipulate input data to induce model misclassification [51,52]. While the DeepFool method iteratively perturbs input images until misclassification occurs, JSMA targets the most influential input features to mislead the model. Developing models that are resistant to such adversarial manipulations is especially critical in high-stakes domains such as medical diagnosis. Robust AI systems hold significant potential for medical imaging by enabling reliable analysis of radiological data under diverse and potentially adversarial conditions. In this context, AI resilience can be viewed as a multidimensional concept encompassing several forms of

robustness. One important dimension is data resilience, which refers to a model's ability to maintain stable performance when confronted with incomplete, imbalanced, or heterogeneous datasets [53].

**3.2.1 Model robustness.** The robustness of the model is the ability to maintain high performance despite variations or perturbations in input data, such as noise, adversarial attacks, or changes in data distribution [54]. A robust model is less sensitive to small, unpredictable changes in input and can generalize well to unseen data. Obtaining robustness often involves regularization techniques, data augmentation, and adversarial training, which enhance the resilience of the model to internal and external variations, ensuring reliable predictions in diverse scenarios [55].

**3.2.2 Adversarial robustness.** The unequal distribution of sensitive features between the control and PD groups causes bias in the dataset. With older people and specific racial groups (such as white patients) being overrepresented in comparison to younger or minority populations, age and race traits are noticeably skewed. Subtle disparities still exist, even if the proportion of genders is more equitable. The ability of a model to continue producing accurate predictions in the face of intentionally altered inputs is known as adversarial robustness [56]. Such robustness is essential in clinical AI systems because even small input perturbations can result in potentially fatal diagnostic errors [57]. By specifically including adversarial examples during training, adversarial training has been acknowledged as one of the best methods for improving generalization and defense against such disturbances. For example, Madry et al. [58] showed how adversarial training greatly increases model resilience under these attacks, and Goodfellow et al. [59] introduced the Fast Gradient Sign Method (FGSM) to create adversarial samples. By maintaining data diversity and privacy, federated learning (FL), which decentralizes model training across devices or institutions without sharing raw data, naturally promotes robustness [59,60]. Recent studies have demonstrated that incorporating adversarial defense mechanisms within FL can improve the robustness and effectiveness of clinical models in addition to their privacy benefits [61]. Resilient FL frameworks in particular have demonstrated enhanced performance in heterogeneous medical datasets and have reduced the risk of centralized data poisoning [62,63].

In addition, auditing and maintaining robustness in healthcare AI depend on Explainable AI (XAI). By showing when a model's predictions rely too heavily on spurious features, techniques like SHAP and LIME can be used to identify vulnerabilities and provide insights into model decision pathways [64]. As highlighted in recent guidelines for reliable medical AI, XAI also facilitates robust model validation by enabling clinicians to confirm whether model behavior is consistent with clinical reasoning [65]. It is becoming more widely accepted that establishing standardized frameworks for robustness evaluation is an essential first step in implementing AI in healthcare. More reliable and equitable AI systems are ensured by benchmarking models against known attack vectors and across fairness-sensitive attributes including gender or race [66].

**3.2.3 Data bias.** In ML, challenges introduced during model development and training can contribute to the emergence of bias. The assumptions, design choices, and implicit cognitive biases of the developers may be reflected in the models they construct. This issue is further exacerbated by the use of inaccurate, incomplete, or biased datasets for training and evaluation [67]. In particular, demographic attributes such as gender, age, and race are often unevenly distributed across classes, leading to systematic biases in the data. Empirical observations indicate that age and race exhibit greater skewness compared to gender-related features. For instance, in both the non-PD and PD cohorts, the majority of patients are 60 years or older, highlighting a pronounced age imbalance in the dataset. It is crucial to consider differences in clinical outcomes and data representation when analyzing gender, age and race-related bias in PD prediction models. For example, if the dataset is not sufficiently balanced, model learning can be biased because women are more likely to experience sadness and anxiety, and men with PD are more likely to report stiffness and sleep disturbances [68]. Furthermore, due to age-related multi-morbidities and unusual symptomatology, older people are often underrepresented in AI datasets, which can lead to biased predictions [69]. The particularly concerning aspect is racial prejudice, as Black and Hispanic patients are underrepresented in study cohorts and often have delayed PD diagnoses, which reduces the generalizability of the model [70,71]. These variations highlight the need for stratified sampling, intersectional equity evaluations, and ongoing model assessments to prevent the perpetuation of systemic bias.

 

### 3.3 Evaluation metrics

Employing a set of indicators, commonly referred to as justice metrics, enables the identification of preprocessing bias in datasets as well as in in-processing models, which may exhibit either unintentional or deliberate favoritism toward certain groups. These indicators, summarized in Table 2, facilitate the detection of bias in the data or model-level and support subsequent mitigation efforts. From a bias identification perspective, this approach highlights instances of unfair treatment of one group relative to another [72]. Bias mitigation refers to a set of techniques aimed at reducing bias in datasets or models used in empirical research. In this context, adversarial evaluation assesses the robustness of a model to perturbed inputs designed to reveal or amplify underlying biases, thereby enhancing the evaluation of model fairness. Under standard evaluation conditions, such methods can expose latent model vulnerabilities. In general, adversarial evaluation provides a practical framework for assessing the extent to which a model is susceptible to adversarial manipulations that may exacerbate its inherent biases [73].

**3.3.1 Bias metrics.** The protected groups, which we denote as $h_i$ and $h_j$, are considered in the analysis. The first category of metrics, known as parity-based metrics, focuses on comparing the expected positive rates, such as $Q\left(\hat{Z} = 1\right)$, across different groups.

**1. Statistical parity difference:** A distance measure is used to determine similarity [74] if findings are comparable between groups and unrelated to the protected property. This is achieved as follows:

$$Q\left(\hat{Z} = 1 \mid h_i\right) = Q\left(\hat{Z} = 1 \mid h_j\right)$$

- $Q(\hat{Z} = 1 \mid h_i)$: This represents the probability that the predicted outcome $\hat{Z}$ equals 1, given the condition or group $h_i$.
- $Q(\hat{Z} = 1 \mid h_j)$: This represents the probability that the predicted outcome $\hat{Z}$ equals 1, given the condition or group $h_j$.

**2. Disparate impact:** Based on confusion matrix measurements, this class of measures spans elements outside the positive rate [60], including the true positive rate (TPR) and true negative rate (TNR). It specifically contrasts the chances of a favorable result for rich and poor groups.

$$\frac{\Pr(Z = 1 \mid D = \text{unprivileged})}{\Pr(Z = 1 \mid D = \text{privileged})}$$

**Table 2. This table shows the error rates and predictive values used in our analysis.** These metrics play a crucial role in evaluating performance.

| Metric | Equation |
|---|---|
| True positive rate (TPR) or sensitivity | $TPR = \frac{TP}{TP+FN}$ |
| True negative rate (TNR) or specificity | $TNR = \frac{TN}{TN+FP}$ |
| False positive rate (FPR) or Type-I error | $FPR = \frac{FP}{FP+TN}$ |
| False negative rate (FNR) or Type-II error | $FNR = \frac{FN}{FN+TP}$ |
| False discovery rate (FDR) | $FDR = \frac{FP}{FP+TP}$ |
| False omission rate (FOR) | $FOR = \frac{FN}{FN+TN}$ |
| Positive predictive value (PPV) | $PPV = \frac{TP}{TP+FP}$ |
| Negative predictive value (NPV) | $NPV = \frac{TN}{TN+FN}$ |

where TP (true positive) denotes correctly identified diseased patients; TN (true negative) denotes correctly identified cognitively normal individuals; FN (false negative) refers to diseased subjects who are incorrectly predicted as healthy by the model; and FP (false positive) refers to healthy individuals who are incorrectly predicted as diseased.

**3. Equal opportunity difference:** This criterion ensures that the false negative rate (FNR), defined as the probability that an instance belonging to the positive class is incorrectly classified as negative, is equal across both protected and unprotected groups. When this condition is satisfied, the classifier is considered to satisfy the fairness requirement [75].

$$Q(Z = 1 \mid d = 1, G = m) = Q(Z = 1 \mid d = 1, G = f)$$

**4. Average absolute odds difference:** The AAOD metric, which calculates bias using the false positive rate and true positive rate, is considered fair when it equals 0. This measure reflects the disparity in performance between groups, where fairness implies equal treatment of both groups.

$$\text{AAOD} = \frac{1}{2}\Big[|FPR_{A=\text{minority}} - FPR_{A=\text{majority}}|$$
$$+ |TPR_{A=\text{minority}} - TPR_{A=\text{majority}}|\Big]$$

Where the group of sensitive quality is $A$, every group is supposed to get an equal share of good results. This method finds it difficult, nevertheless, to handle situations when people belong to many protected categories. Giving one group justice priority could jeopardize fairness for another.

3.3.2 **Group fairness.** Group fairness is the principle that individuals from different demographic groups such as gender, race, or age, should, on average, receive comparable model outcomes. Parity-based measures operationalize this concept by quantifying differences in prediction results between protected groups. As shown in Table 2, common metrics include statistical parity difference (SPD), equal opportunity difference (EOD), and average absolute odds Difference (AAOD) [76]. SPD measures disparities in the probability of favorable outcomes (for example, a correct PD diagnosis) across subgroups, while EOD assesses differences in true positive rates to ensure equal access to beneficial interventions. Although confusion-matrix metrics such as true positive rate (TPR), false positive rate (FPR), and positive predictive value (PPV) underpin these measures, they are only applied within fairness formulations when broken down by group membership. For instance, AAOD captures fairness violations by averaging the absolute differences in group-specific TPR and FPR. Accordingly, this study reports both individual performance metrics and aggregated group-fairness measures to present a comprehensive view of model fairness [77].

## 4 Proposed model

Fig 2 illustrates the proposed framework for the detection of PD, which adopts a fairness-aware design to systematically address algorithmic bias. In this study, the Parkinson's Progression Markers Initiative (PPMI) database is used to extract baseline and multimodal data, including clinical, imaging, and biospecimen records. The preprocessing pipeline comprises feature selection, using techniques such as recursive feature elimination to identify informative predictors, and data normalization, employing z-score standardization to ensure consistent feature scaling. Model optimization is performed through hyperparameter tuning guided by cross-validation, with the objective of improving predictive performance and generalizability. To account for potential biases, an initial discriminatory data unit testing phase is performed to detect disparate impacts across demographic subgroups, using fairness metrics such as demographic parity and equal opportunity. Subsequently, bias mitigation strategies are applied through reweighting techniques and group-specific adjustments informed by fairness-aware criteria. Finally, the performance of classifiers, including random forests and gradient boosting models, is evaluated by comparing the results obtained from the original and bias-adjusted datasets. Fig 3 presents the experimental workflow adopted in this study.

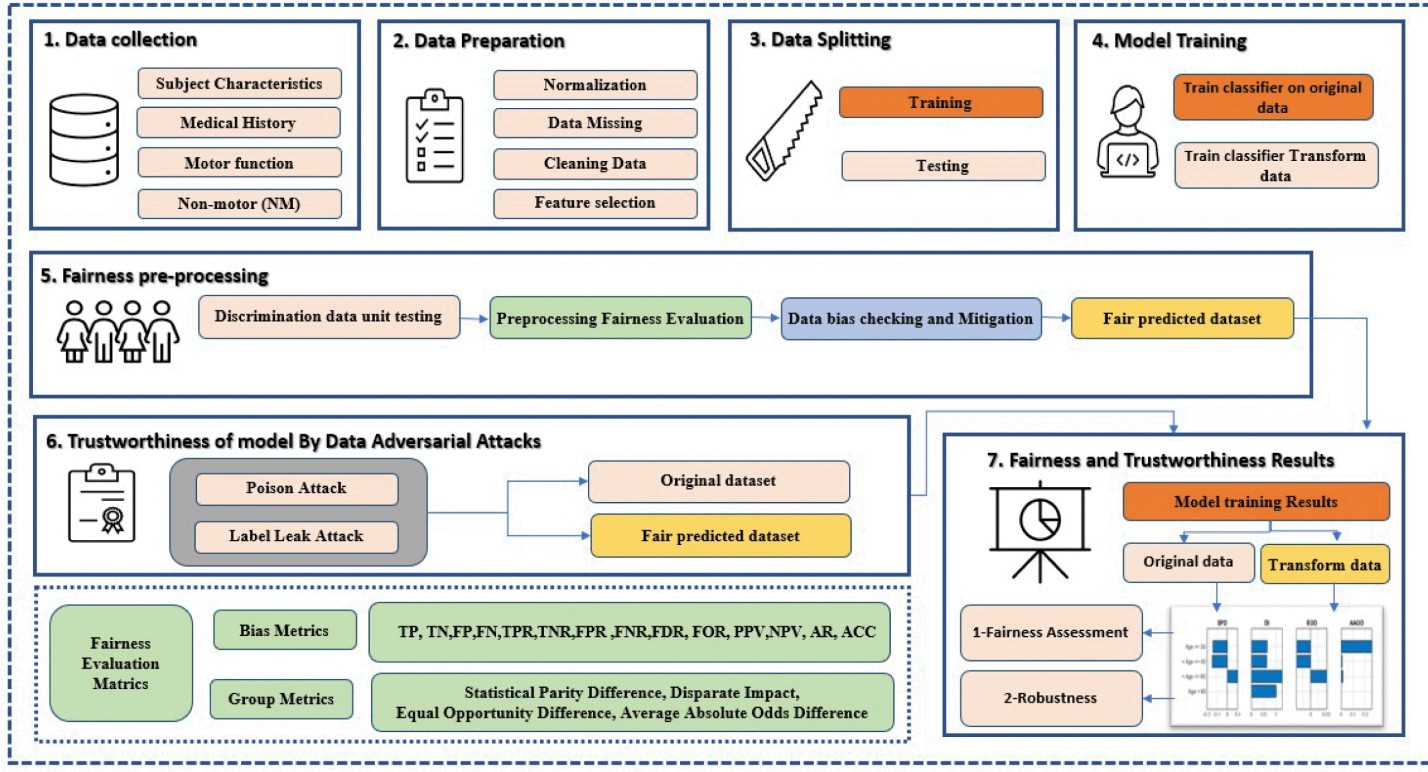

**Fig 2**. Proposed framework for the detection of PD.

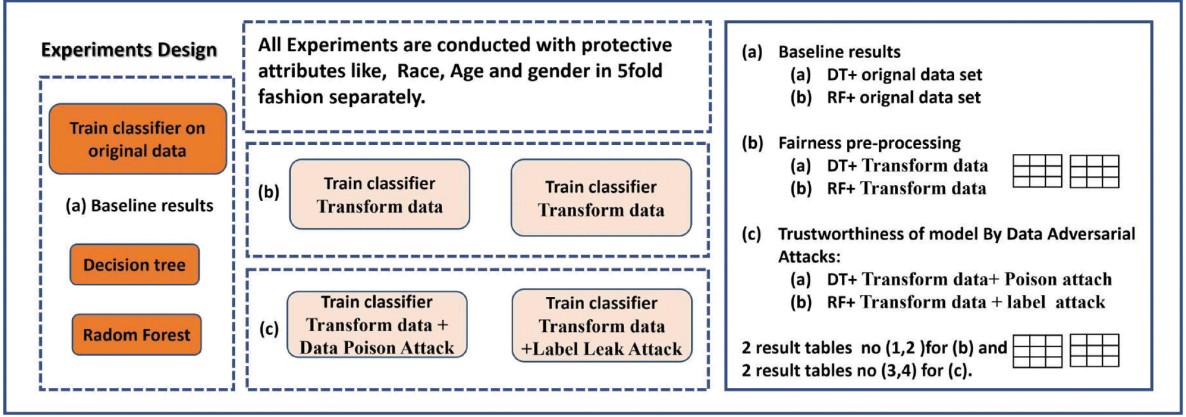

**Fig 3**. The experimental route map of the proposed framework.

## 4.1 Data preparation

We performed a series of preprocessing steps on the training data prior to training and evaluating the proposed model. The initial and most critical step involved identifying the most informative modalities for PD detection. Subsequently, the statistical significance of the selected modalities at baseline was assessed, with each undergoing multiple preprocessing

procedures to enhance data quality. Their discriminative capacity for evaluating PD severity was also examined. Details regarding the dataset and the selected features are provided in [61].

**4.1.1 Data collection.** The Parkinson's Progression Markers Initiative (PPMI) dataset is a publicly available, longitudinal, multicenter cohort study designed to identify biomarkers of PD progression. Globally tracking PD cases and including data from several nations, the PPMI database [62] covers patient clinical data rather brilliantly. As part of the PPMI dataset, it includes each patient's medical records, laboratory results, motor and non-motor test scores, bio-samples, and personal information [67]. The study divides the dataset into numerous data groups with individual traits. The investigation covers the fundamental modalities, including motor and non-motor, as well as subject characteristics and medical history. The De Novo PD cohort consists of individuals diagnosed within two years of enrollment who have not commenced dopaminergic treatment. Healthy Controls (HC) are meticulously matched to PD participants based on age and sex and must possess no history of neurological dysfunction, no first-degree relatives with PD, and a Montreal Cognitive Assessment (MoCA) score of no less than 26. The prodromal cohort comprises individuals aged 60 years or older who display high-risk indicators for Parkinson's disease, such as REM sleep behavior disorder (RBD), yet do not fulfill clinical diagnostic criteria. Finally, the genetic cohort consists of both symptomatic and asymptomatic carriers of pathogenic mutations in the LRRK2, GBA, and SNCA genes. To maintain cohort integrity, stringent exclusion criteria are enforced: PD participants must not exhibit signs of atypical parkinsonism, and controls must undergo comprehensive cognitive and motor assessments. The PD dataset is provided by the PPMI as a time series, which reflects patient visits in Table 3. For their six regular visits, patients follow a 12-month interval schedule: Baseline (BL) visit 1 at month 12, visit 2 at month 24, visit 3 at month 36, visit 4 at month 48, and visit 5 at month 60 as shown in Fig 4. Using the H&Y scale and a variable NHY, patients are categorized. The dataset baseline information is that all, 1059 patients are part of it. Using the current standard stages, we use the data preparation techniques: mean imputation fills in missing values, the minimum-maximum scaler function normalizes, and the recursive elimination strategy helps us to choose features. With gender and racial attributes translated into a 0/1 numerical data type, the goal variable is a binary variable where 1 denotes Parkinson's patients (PAT n = 648) and HC (healthy control n = 434). Concerning sensitive factors, age and race were categorized to enhance fairness assessment across subgroups while reducing privacy concerns. In our analysis, we categorized age into two bins to align with clinical stratifications commonly used in PD research [78,79] and to ensure sufficient sample size per subgroup for meaningful fairness evaluation. Similarly, race was binarized into "White" and "Non-White" to preserve statistical power and participant anonymity, given the small counts in minority subgroups. Gender was retained as

**Table 3. Overview of the PPMI database for Parkinson's disease, including patient demographic information, laboratory results, clinical records, and motor and non-motor assessment outcomes.**

| Characteristic | Visit 1 | | | Visit 6 | | |
|---|---|---|---|---|---|---|
| | PAT (n=648) | HC (n=434) | SD | PAT (n=673) | HC (n=410) | SD |
| Male (%) | 391 (60.33%) | 202 (47.0%) | – | 408 (60.62%) | 187 (45.6%) | – |
| Age at inclusion (Mean) | 64.23 | 62.83 | 0.99 | 64.23 | 62.83 | 0.99 |
| Education (Mean) | 15.47 | 16.95 | 1.05 | 15.47 | 16.95 | 1.05 |
| UPSIT (Smell Identification Test) | 5.66 | 7.80 | 1.51 | 5.65 | 7.97 | 1.64 |
| Semantic Fluency (animals, vegetables, fruits) | 28.43 | 30.36 | 1.36 | 29.00 | 30.09 | 0.77 |
| REM Sleep Behavior Questionnaire | 0.26 | 0.22 | 0.03 | 0.30 | 0.18 | 0.08 |
| Epworth Sleepiness Scale (ESS) | 0.82 | 0.72 | 0.07 | 0.94 | 0.68 | 0.18 |
| Montreal Cognitive Assessment (MoCA) | 1.07 | 1.03 | 0.03 | 0.99 | 1.02 | 0.02 |
| Geriatric Depression Scale (GDS-15) | 0.36 | 0.34 | 0.01 | 0.38 | 0.34 | 0.03 |
| UPDRS I | 0.57 | 0.44 | 0.09 | 0.82 | 0.47 | 0.25 |
| UPDRS II | 0.49 | 0.00 | 0.35 | 0.81 | 0.10 | 0.50 |
| UPDRS III | 3.30 | 2.90 | 0.28 | 3.28 | 2.90 | 0.27 |
| Hoehn & Yahr stage | 1.66 | 0.00 | 1.17 | 2.00 | 0.00 | 1.41 |
| Schwab & England (%) | 90.40 | 96.90 | 4.60 | 83.63 | 96.18 | 8.87 |
| Fatigue Severity Scale (FSS) | 0.78 | 0.53 | 0.18 | 0.89 | 0.47 | 0.30 |

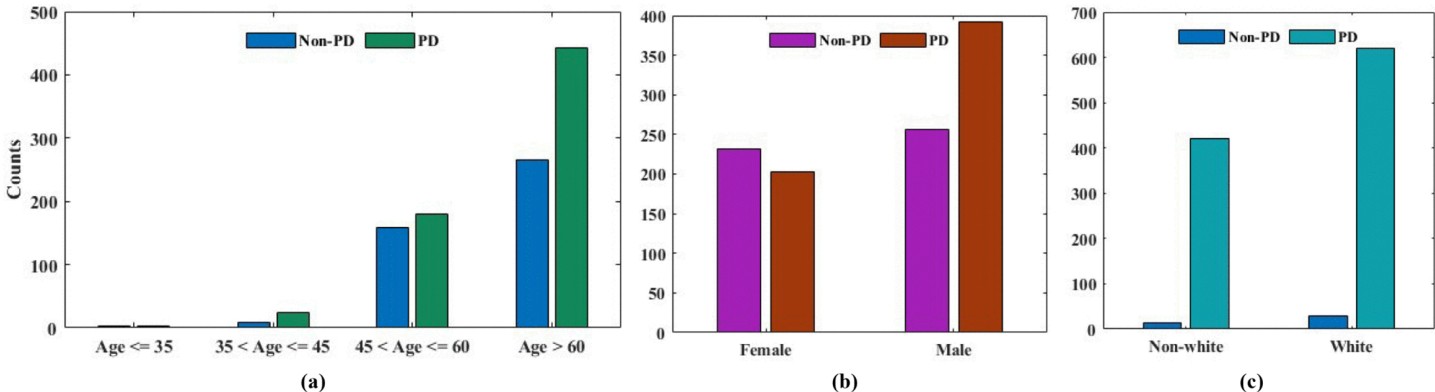

**Fig 4**. Illustrates how PD patients and non-PD patients differ in terms of (a) age, (b) gender, and (c) race.

a binary variable (male/female) per PPMI coding. Importantly, these sensitive attributes (age, gender, and race) were excluded from model training features to prevent direct bias amplification [78], but were retained for post-hoc fairness auditing across demographic subgroups. This design enables us to assess whether the model's predictions exhibit disparities despite not using protected attributes as inputs, thus revealing latent biases embedded in clinical and biomarker data.

**4.1.2 Data cleaning.** It uses data cleaning methods to make the dataset match up with other related datasets in the process. Fixing errors involves formatting data, getting rid of duplicates, and turning numbers saved as text back into numbers [80].

**4.1.3 Missing data.** Removes a feature if it has more than 30% missing data. To fill in the blanks, researchers have utilized a variety of methods, such as forward and backward filling. In our case, we employed the median to fill in continuous (numerical) traits and the mode method to fill in categorical data.

**4.1.4 Data normalization.** Data normalization is an essential step in the data preprocessing pipeline. Normalization, also known as standardization or feature scaling, is the process of making data without any dimensions and with similar patterns [67,72]. Numerous studies suggested using the min-max normalization approach to make the dataset more uniform across the range [0,1] [81]. These steps involve rescaling the data so that the feature values stay within this range.

Min-Max normalization is given by the following formula:

$$y' = \frac{y - y_{\min}}{y_{\max} - y_{\min}} \times (d - c) + c$$

where $y_{\max}$ and $y_{\min}$ denote the maximum and minimum values, respectively. This normalization technique scales the values within the interval $[c,d]$, transforming the original value $y$ into the normalized value $y'$.

**4.1.5 Data splitting.** The dataset was partitioned into training and testing subsets, with 80% used for model training and 20% retained for evaluation. The dataset exhibits a moderate degree of class imbalance, consisting of 648 PD cases and 434 healthy controls (HC), corresponding to an approximate PD-to-HC ratio of 3:2. This imbalance may bias model predictions toward the majority class, thereby inflating overall accuracy while masking suboptimal performance on the minority class. To mitigate this effect, five-fold stratified cross-validation was employed to preserve class distributions across training and validation partitions. Model performance was assessed using balanced evaluation metrics, including accuracy, recall, F1-score, and group-specific fairness indicators (e.g., *TPR* and *FNR*), in order to ensure sensitivity to minority-class outcomes.

**4.1.6 Feature selection.** Feature selection constitutes a central component of the most ML based studies aimed at predictive modeling [82]. The primary objective is to identify a minimal subset of input features that preserves high predictive performance while maintaining clinical relevance. Reducing the dimensionality of the feature space mitigates overfitting and eliminates redundant or irrelevant variables, allowing the classifier to focus on the most informative predictors [36].In this study, sensitive attributes, including age, gender, and race, were removed from the training feature set to prevent bias amplification, as these variables are used more appropriately for fairness evaluation rather than prediction. Incorporating such attributes can introduce discriminatory effects or confounding relationships that compromise model equity [83]. Consequently, the feature selection strategy was determined based on extensive empirical evaluation. Recursive Boosting and Stability and Uncorrelated Local Optima-based Variable Selection (SULOV) were employed to identify the most relevant features for model construction. By leveraging iterative XGBoost-based importance ranking in conjunction with SULOV, the proposed approach facilitates the selection of a compact and informative feature subset that enhances both predictive accuracy and model interpretability.

For instance, the dataset used in this study is represented as $\mathcal{D} = \{(z_n, w_n) : n = 1, \dots, p, \ z_n \in \mathbb{R}^q, \ w_n \in \mathbb{R}\}$, with $p$ samples and $q$ features. The model's prediction is $\hat{w}_n = \sum_{l=1}^{r} G_l(z_n)$, where $G_l$ is the $l$-th regression tree. The prediction score given to the $n$-th sample by the $l$-th tree is $G_l(z_n)$. By minimizing the objective function described below, we can learn the set of functions $G_l$ in the regression tree model:

$$J = \sum_{n=1}^{p} M(w_n, \hat{w}_n) + \sum_{l=1}^{r} \Lambda(G_l)$$

$\Lambda(G_l)$ is the regularizing term for the $l$-th regression tree, which helps to prevent overfitting. Here, $M(w_n, \hat{w}_n)$ is the loss function, measuring the difference between the actual value $w_n$ and the projected value $\hat{w}_n$.

We can efficiently find the most significant features by applying recursive XGBoost and SULOV, guaranteeing that the model stays accurate and interpretable.

## 4.2 Optimization of baseline ML models

We evaluated supervised ML models to predict the progression of PD in order to construct a prognostic framework for the disease advancement. These ML techniques include Random Forest (RF) [84] and Decision Tree (DT) [85].These models are well suited for binary classification tasks and are capable of leveraging multiple combinations of modality. We optimized model hyperparameters using a stratified 5-fold cross-validation and grid search technique, and selected the configurations yielding the lowest training loss. The trained models with optimal hyperparameter settings were subsequently used for PD detection to ensure reliable predictive performance. The initial application of these techniques focused primarily on accuracy and predictive ability, without incorporating fairness constraints. This stage served to reveal potential model biases and to establish a baseline for subsequent fairness-aware comparisons. We utilized a decision-tree-based simulation framework for analysis and evaluation. In addition, we included a baseline DL algorithm to enable direct comparison under identical data splits and preprocessing conditions. This baseline demonstrates that a standard MLP can achieve competitive predictive performance while still exhibiting fairness disparities, thus supporting the need for bias mitigation strategies irrespective of model class [86].

**4.2.1 Decision tree.** Based on user-defined criteria for the target variable, the predicted outcome is inferred using a DT classifier [87]. Classification trees are hierarchical structures in which internal nodes represent decision functions and terminal nodes (leaves) correspond to distinct subsets of the target variable, referred to as classes. Decision Trees are widely adopted MI algorithms due to their conceptual simplicity, interpretability, and ease of implementation. Most DT-based algorithms employ a top-down recursive partitioning strategy, in which a feature is elected at each node that maximizes the separation of the data into homogeneous subsets. The selection of an optimal splitting criterion is guided

by a range of statistical measures that quantify the purity of the resulting partitions. These measures assess the distribution of class labels within each subset of the dependent variable. By applying this process recursively to all subgroups, the average reduction in impurities provides an estimate of the quality of the split [88]. Both entropy-based information gain and Gini impurity are commonly used splitting criteria, and their mathematical formulations are given as follows:

$$\text{Entropy}: T(z) = -\sum_{i=1}^{p} m(z_i) \log_2 m(z_i)$$

$$\text{Gini}(H) = 1 - \sum_{i=1}^{q} m_i^2$$

**4.2.2 Random forest.** Breiman introduced the RF algorithm as an ensemble learning approach that constructs multiple independent base learners and aggregates their predictions to improve overall performance. The fundamental principle of this method is to reduce variance and improve generalization by combining the outputs of a collection of weak learners. RF incorporates a variant of the traditional bagging (bootstrap aggregating) technique, in which each individual classifier is trained on a bootstrapped sample drawn from the original training dataset. Unlike a conventional decision tree algorithms, which evaluate all available features in each split, RF selects a random subset of features at every node when determining the optimal split. This random feature selection mechanism promotes model diversity and reduces correlation among individual trees. In addition to its strong predictive performance in high-dimensional feature spaces, RF has also been shown to perform robustly in settings with relatively low-dimensional feature representations. Formally, let $\mathcal{X}$ denote the training dataset, and $N$ represent the total number of trees in the ensemble. A set of classification or regression trees $\{T_n\}_{n=1}^{N}$ is constructed using bootstrapped samples from $\mathcal{X}$. For a given unseen input sample $x^*$, the final prediction is obtained by averaging the outputs of all individual trees in the ensemble, which can be expressed as:

$$\hat{T}(x^*) = \frac{1}{N} \sum_{n=1}^{N} T_n(x^*) \tag{1}$$

**4.2.3 Multi-Layer perceptron.** We included a two-hidden-layer multilayer perceptron (MLP) as a representative deep learning baseline model for binary classification. By composing affine transformations with ReLU nonlinearities, the MLP captures nonlinear feature interactions and is trained in an end-to-end to manner using a binary cross-entropy loss. Regularization was applied via batch normalization, dropout, weight decay, and early stopping to improve generalization and to enable a reproducible comparison with the classical baselines [89]. However, the MLP performance to data availability and hyperparameter tuning is typically less interpretable than tree-based models and requires feature scaling and controlled random seeding for reliable reproducibility. A general l-layer MLP forward pass can be represented as:

$$\begin{aligned}
\mathbf{h}^{(0)} &= \mathbf{x} \in \mathbb{R}^d, \\
\mathbf{a}^{(\ell)} &= \mathbf{W}^{(\ell)} \mathbf{h}^{(\ell-1)} + \mathbf{b}^{(\ell)}, \quad \ell = 1, \dots, L, \\
\mathbf{h}^{(\ell)} &= \phi\big(\mathbf{a}^{(\ell)}\big), \quad \ell = 1, \dots, L-1, \\
\hat{\mathbf{y}} &= g\big(\mathbf{a}^{(L)}\big).
\end{aligned}$$

**Hidden-layer nonlinearity.**

$$\phi : \mathbb{R} \to \mathbb{R}, \qquad \phi(z) = \text{ReLU}(z) = \max(0, z).$$

**Output mapping.**

$$g : \mathbb{R}^m \to \mathbb{R}^C, \qquad g(z) = \begin{cases} \text{id}(z) & \text{(regression)}, \\ \sigma(z) = \dfrac{1}{1 + e^{-z}} & \text{(binary classification)}, \\ \text{softmax}(z)_k = \dfrac{e^{z_k}}{\sum_{j=1}^{C} e^{z_j}} & \text{(multiclass classification)}. \end{cases}$$

**Deep baseline configuration.** We used an MLP with architecture $d \to 256 \to 128 \to 1$, ReLU activations, batch normalization in hidden layers, and dropout with $p = 0.20$. The trainable weights were initialized using the He initializer. The network was trained with `BCEWithLogitsLoss` loss funtion. Optimization was performed using Adam ($\text{lr} = 1 \times 10^{-3}$, weight_decay $= 1 \times 10^{-5}$), together with `ReduceLROnPlateau` (factor 0.5, patience 3). Training was run for up to 100 epochs with early stopping based on validation AUROC (patience 10, restoring the best checkpoint) and a batch size of 16. Numerical features were preprocessed using a standardized z-score normalization approach, and categorical variables were converted to a one-hot encoding format. All transformations were initially applied only to the training fold and then applied to the validation and test folds. The decision threshold was selected on the validation split to maximize the F1-score and then applied to the test split. We reported Accuracy, F1-score, AUROC, and fairness gaps (SPD and EOD) as mean $\pm$ 95% confidence intervals over 10-fold cross-validation. This deep baseline provides a direct check that fairness interventions remain effective beyond model family differences, supporting the use of the proposed mitigation strategy in both classical and DL models.

## 4.3 Fairness preprocessing

Pre-processing, in-processing, and post-processing approaches are commonly used to mitigate bias in machine learning pipelines. In particular, pre-processing methods aim to reduce biases present in the training data, thereby limiting the propagation of bias through subsequent modeling stages. A probabilistic formulation of data preparation has been proposed to address dataset-level bias. Under group-fairness, individual-distortion, and data-integrity constraints, optimized pre-processing applies probabilistic modifications to both features and labels to produce a less biased representation of the data. Robust optimization in this setting is typically guided by three objectives: minimizing distortion to individual samples, controlling discriminatory effects across groups, and preserving predictive utility while learning the data transformation [90].

## 4.4 Adversarial attacks

This section highlights a clear overview of the structure of poison and label leak attacks and their possible consequences on the fairness of ML models. These conversations provide a basis for future research on mitigation techniques to guaranty reliable results [91,92].

**4.4.1 Poison attack.** A poisoning attack refers to the intentional injection of adversarially crafted samples into the training set, with the goal of biasing the learning process and compromising model integrity [93]. Consider a setting in which predictions are intended to be independent of a sensitive attribute $t$. Under label leakage, however, the model may learn a mapping $f$ that directly (or indirectly) uses $t$, yielding

$$\hat{z} = f(z, t; \theta).$$

In the ideal case, the model would instead learn a function $k$ that relies only on the non-sensitive features $z$, such that

$$\hat{z} = k(z; \theta),$$

and the prediction $\hat{z}$ is not influenced by $t$. The disparity induced by dependence on $t$ can be quantified by comparing expected prediction errors across groups defined by $t$:

$$\Delta_{\text{error}} = |\mathbb{E}[M(k(z;\theta),z) \mid t=1] - \mathbb{E}[M(k(z;\theta),z) \mid t=0]|,$$

where $M(\cdot)$ denotes the loss function and $\mathbb{E}[\cdot]$ denotes expectation.

**4.4.2 Label leak attack.** Label leakage is inadvertent learning of the model to depend on labels connected with a sensitive attribute and results in biased forecasts. One may express a label leak attack numerically as follows: let the forecast and a sensitive attribute $s$ have no relationship. But label leaking causes the model to learn a function $g$.

$$\hat{y} = g(x,s;\theta)$$

The model should ideally learn a function $h$ that $s$ has no influence on the prediction $\hat{y}$, and:

$$\hat{y} = h(x;\theta)$$

The difference in prediction errors among groups defined by $s$ allows one to measure the label leakage:

$$\Delta_{\text{error}} = |E[L(h(x;\theta),y) \mid s=1] - E[L(h(x;\theta),y) \mid s=0]|$$

where the expected is represented by $E$.

## 4.5 Experimental setup

In this study, Python version 3.8 was used for conducting experiments, with Scikit-learn for conventional ML implementations and PyTorch framework for DL based model training and evaluation. An MSI Pulse 17 B13V laptop with an Intel Core i9 CPU and 48GB of RAM was used for model training, adverserial attack simulations, and hyperparameter optimization.

## 5 Results

In our experiments, we evaluated DT and RF classifiers under two conditions: (i) standard training without fairness mitigation, and (ii) training with an optimized pre-processing fairness mitigation method. The performance of the model was examined both overall and with respect to sensitive attributes (age, sex, and race) using a combination of individual- and group-level bias indicators. The dataset was divided into training and testing subsets (80%/20%). Teh hyperparameters were tuned using a grid search technique, and each model was trained on the full dataset containing all demographic groups. Each experiment was repeated five times; the results were reported as the mean $\pm$ standard deviation (SD). Consequently, two result tables were provided for each classifier. The hyperparameters were summarized in Table 4. In addition, a baseline for MLP was reported in Sect 4.2.3, with implementation details provided in Table 5.

All reported metrics were verified to lie within their expected ranges. Specifically, performance measures (*FPR*, *FNR*, *TPR*, and *TNR*) were constrain

### 5.1 Fairness assessment

**Hypothesis 01**: Due to bias reduction, data biases and fairness evaluation of the decision tree with baseline will display dropped outcomes.

**Table 4. Final chosen hyperparameters for machine learning models and adversarial attacks (this work).** Seeds follow `seed = base`.

| Component | Hyperparameter | Final value/Description |
|---|---|---|
| DT | Max depth | **10** |
| | Min samples per leaf | **2** |
| | Splitting criterion | **Gini impurity** |
| | Class weight | **None** (default) |
| | Random seed | **4765416** |
| RF | Number of trees (estimators) | **200** |
| | Max depth | **12** |
| | Min samples per split | **4** |
| | Max features per split | `sqrt` |
| | Bootstrap | **True** |
| | Class weight | **None** (default) |
| | Random seed | **4765416** |
| Label poisoning (train-only) | Attack type | **Label flipping (indiscriminate)** |
| | Poisoning rate | **5%** of **training** labels (main); sensitivity at {0,5,10,15,20}% |
| | Stratification | **class** × **sensitive_group** (uniform selection within strata) |
| | Flip rule (binary) | $y \leftarrow 1 - y$ |
| | Apply to | **Training folds only**; validation/test remain clean |
| | Random seed | **4765416** |
| | Logged checks | Realized rate, flips per stratum, priors before/after |
| Label leakage (proxy feature) | Leak feature | `leak_s` (binary) |
| | Leakage degree | $\alpha = 0.90$ agreement with $s$ (flip to 1–$s$ with prob 0.10) |
| | Sensitive attribute | $s \in \{0, 1\}$ |
| | Apply to | **Train + validation + test** (systemic leakage) |
| | Pipeline position | After preprocessing, before feature dropping; `leak_s` retained |
| | Random seed | **4765416** |
| | Logged checks | Proxy accuracy $P(\text{leak\_s}=s) \approx 0.90$ |

**Table 5. MLP (DL baseline) with and without adversarial robustness (adversarial debiasing).** Entries are mean ± 95% CI over 10-fold CV. Higher is better for Accuracy/F1/AUROC; lower is better for fairness gaps (SPD, EOD).

| Evaluation metrics | Race | | Age | | Gender | |
|---|---|---|---|---|---|---|
| | **Without** | **With** | **Without** | **With** | **Without** | **With** |
| Accuracy (%) | **76.3** ± 1.5 | **74.9** ± 1.6 | **75.1** ± 1.6 | **74.4** ± 1.5 | **76.8** ± 1.4 | **75.6** ± 1.5 |
| F1-score (%) | **72.7** ± 1.8 | **71.1** ± 1.7 | **73.1** ± 1.7 | **71.8** ± 1.8 | **73.6** ± 1.6 | **72.2** ± 1.7 |
| AUROC (%) | **79.2** ± 1.1 | **77.3** ± 1.4 | **77.6** ± 1.3 | **76.8** ± 1.2 | **78.4** ± 1.2 | **77.0** ± 1.3 |
| SPD |Δ| (%) | **9.4** ± 2.1 | **3.5** ± 1.2 | **8.6** ± 1.9 | **4.2** ± 1.1 | **9.0** ± 2.0 | **3.9** ± 1.2 |
| EOD |Δ| (%) | **11.1** ± 2.4 | **5.0** ± 1.4 | **10.5** ± 2.2 | **6.1** ± 1.5 | **11.8** ± 2.3 | **5.7** ± 1.4 |

[1] SPD : statistical parity difference, EOD : equal opportunity difference, "With" uses adversarial debiasing (in-processing) on the same MLP, "Without" is the baseline MLP.

**Hypothesis 02**: Due to bias reduction, data biases, and the fairness assessment of Random Forest with baseline, it will display decreased results.

**Hypothesis 03**: The findings and the efficiency of bias mitigation will be affected by the complexity of the model computation.

**5.1.1 Data biases and fairness evaluation of decision tree.** Table 6 shows the model performance metrics together with variations in certain bias and group measures, both with and without the use of the improved preprocessing fairness mitigating strategy. This table shows the approach followed in the DT model and shows how the fairness strategy reduces prejudices about gender, age, and color.

After fairness mitigation was put into effect, model performance clearly changed. Accuracy dropped across all demographic factors, for example, from 82.12% to 74.56% for race, from 80.32% to 76.24% for age, and from 85.15% to

**Table 6**. **This table provides the five-fold performance difference between the DT classifier's performance with and without the enhanced pre-processing fairness mitigation technique.** We provided two types of fairness evaluation: group metrics (*TPR, TNR, FPR, FNR, FDR, FOR, PPV, and NPV*) and bias measures (*SPD, DI, EOD, and AAOD*).

| Evaluation metrics | Race | | Age | | Gender | |
|---|---|---|---|---|---|---|
| | Without | With | Without | With | Without | With |
| Accuracy | $82.12 \pm 0.07$ | $74.56 \pm 0.05$ | $80.32 \pm 0.09$ | $76.24 \pm 0.08$ | $85.15 \pm 0.03$ | $77.08 \pm 0.07$ |
| Precision | $70.10 \pm 0.05$ | $71.12 \pm 0.03$ | $70.04 \pm 0.07$ | $73.02 \pm 0.06$ | $77.10 \pm 0.04$ | $73.05 \pm 0.05$ |
| Recall | $70.48 \pm 0.04$ | $71.35 \pm 0.02$ | $69.20 \pm 0.08$ | $72.12 \pm 0.06$ | $74.02 \pm 0.03$ | $72.12 \pm 0.05$ |
| F1-score | $66.02 \pm 0.07$ | $66.45 \pm 0.04$ | $63.15 \pm 0.10$ | $70.65 \pm 0.08$ | $73.40 \pm 0.03$ | $71.02 \pm 0.07$ |
| SPD | −0.4512 | −0.0058 | −0.29 | −0.26 | −0.03 | −0.18 |
| DI | 0.3887 | 0.9942 | 0.49 | 0.55 | 0.96 | 0.70 |
| EOD | −0.2981 | −0.017 | −0.19 | −0.35 | 0.06 | −0.12 |
| AAOD | −0.3523 | −0.002 | −0.42 | −0.20 | −0.02 | −0.22 |
| TPR | 0.05 | 0.04 | 0.06 | 0.05 | 0.02 | 0.05 |
| TNR | 0.03 | 0.02 | 0.03 | 0.02 | 0.03 | 0.02 |
| FPR | 0.97 | 0.98 | 0.97 | 0.99 | 0.98 | 0.99 |
| FNR | 0.96 | 0.97 | 0.95 | 0.97 | 0.96 | 0.96 |
| FDR | 0.96 | 0.96 | 0.95 | 0.97 | 0.95 | 0.96 |
| FOR | 0.98 | 0.99 | 0.97 | 0.98 | 0.98 | 0.99 |
| PPV | 0.05 | 0.03 | 0.06 | 0.03 | 0.05 | 0.05 |
| NPV | 0.02 | 0.01 | 0.02 | 0.01 | 0.02 | 0.01 |

[1] SPD : statistical parity difference, DI : disparate impact, EOD : equal opportunity difference, AAOD : average absolute odds difference, TPR : true positive rate, TNR : true negative rate, FPR : false positive rate, FNR : false negative rate, FDR : false discovery rate, FOR : false omission rate, PPV : positive prediction value, and NPV : negative predictive value.

77.08% for gender. These modifications show the compromise between the decrease of systematic data biases and general model performance. Precision, recall, and F1-score measures, however, indicated only minor changes that highlighted the balanced character of forecasts following mitigation.

Additionally, showing notable gains were the measures of fairness bias: *SPD, DI, EOD, and AAOD*. For the racial characteristic,*SPD* declined from −0.4512 to −0.0058, but *DI* rose from 0.3887 to 0.9942, therefore showing more fair treatment for the groups. In the same vein, *AAOD* dropped significantly from −0.3523 to −0.002, therefore confirming the success of the fairness approach. For age and gender, there was a clear drop in prejudice measures showing advancement toward justice, even if the changes were less noticeable than in race.

While the false rate measures (*FPR, FNR, and FDR*) were always high, group metrics including *TPR, TNR, and PPV* exhibited modest changes, thereby highlighting the ongoing difficulties of guaranteeing justice in severely unbalanced datasets. The false omission rate (*FOR*) stayed high across all groups despite the mitigating measures, which reflects the difficulties in correcting strongly ingrained differences in prediction mistakes.

Figs 5 and 6 highlight the outcomes even further. Following the mitigating strategy, *SPD, EOD, and AAOD* metrics reached zero for the race characteristic, while DI neared one, thereby showing significant fairness improvements. The results were varied for age and gender traits, most likely due to underlying dataset discrepancies. Specifically, the gender characteristic displayed opposite bias patterns, whereby group-level measurements indicated differences that remained following mitigation.

Although the pretreatment fairness mitigating strategy significantly reduced bias across several demographic variables, it also brought a trade-off with general model accuracy and predictive dependability. These results emphasize the need for ongoing research to maximize fairness strategies while preserving high model performance.

**5.1.2 Data biases and fairness evaluation of random forest.** We evaluated the fairness of the *DT* and *RF* models. Since both of these models belong to the tree-based classifier family, researchers in the PD domain find them similar and often use them. The *RF* model's performance is described in Table 7, in which the top row presents accuracy matrices

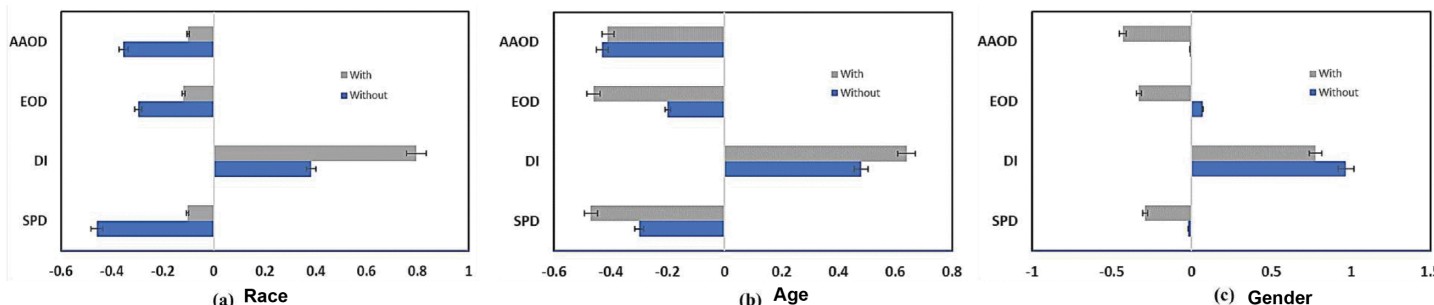

**Fig 5**. Improved preprocessing fairness mitigation strategy, we can see reduced bias disparities across race, age, and gender features in the Decision Tree model, leading to fairer and more equitable outcomes across these protected groups.

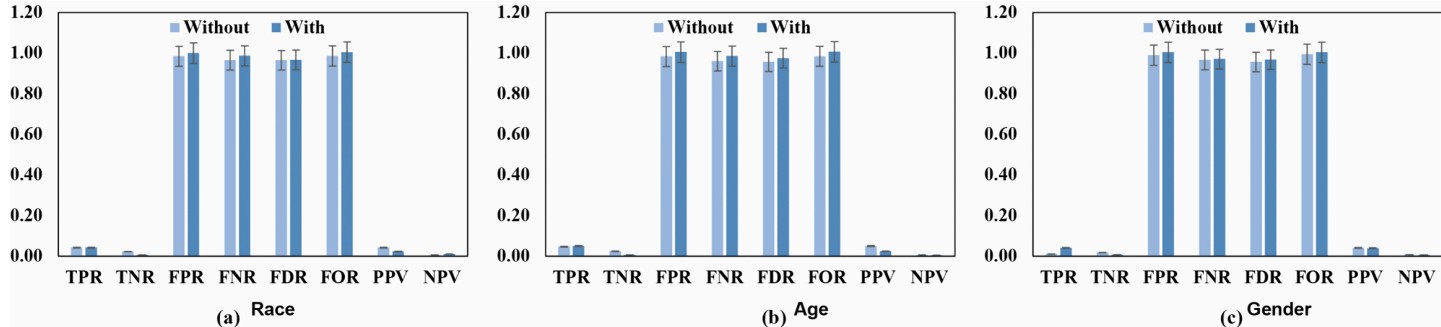

**Fig 6**. For each of the sensitive characteristics, we can provide a brief summary of the DT ML classifier's group measure (bias or performance disparities) both with and without the use of the optimal pretreatment fairness mitigation strategy: (a) race, (b) age, and (c) gender.

**Table 7**. **Report the difference in the RF classifier's five-fold performance between using and not using the improved preprocessing fairness mitigation approach.** Bias measures (*SPD, DI, EOD, and AAOD*) and group metrics (*TPR, TNR, FPR, FNR, FDR, FOR, PPV, and NPV*) are two categories of fairness evaluation that are presented.

| Evaluation metrics | Race | | Age | | Gender | |
|---|---|---|---|---|---|---|
| | **Without** | **With** | **Without** | **With** | **Without** | **With** |
| Accuracy | 84.85 ± 0.04 | 73.52 ± 0.03 | 79.92 ± 0.06 | 75.10 ± 0.04 | 84.12 ± 0.03 | 76.32 ± 0.05 |
| Precision | 73.15 ± 0.05 | 70.76 ± 0.03 | 74.78 ± 0.04 | 72.83 ± 0.05 | 71.89 ± 0.06 | 69.33 ± 0.04 |
| Recall | 72.86 ± 0.06 | 71.02 ± 0.03 | 73.24 ± 0.03 | 70.92 ± 0.04 | 70.41 ± 0.05 | 68.11 ± 0.03 |
| F1-score | 71.98 ± 0.04 | 64.80 ± 0.05 | 73.05 ± 0.04 | 69.52 ± 0.04 | 69.92 ± 0.05 | 67.23 ± 0.04 |
| SPD | −0.38 | −0.05 | −0.28 | −0.18 | −0.15 | −0.08 |
| DI | 0.31 | 0.83 | 0.52 | 0.68 | 0.62 | 0.79 |
| EOD | −0.40 | −0.03 | −0.37 | −0.27 | −0.05 | −0.12 |
| AAOD | −0.35 | −0.04 | −0.22 | −0.09 | −0.08 | −0.10 |
| TPR | 0.03 | 0.04 | 0.06 | 0.07 | 0.04 | 0.05 |
| TNR | 0.01 | 0.03 | 0.01 | 0.02 | 0.02 | 0.04 |
| FPR | 0.98 | 0.99 | 0.98 | 1.00 | 0.98 | 0.99 |
| FNR | 0.97 | 0.98 | 0.96 | 0.97 | 0.97 | 0.98 |
| FDR | 0.96 | 0.97 | 0.96 | 0.98 | 0.96 | 0.97 |
| FOR | 0.99 | 1.00 | 0.99 | 1.00 | 0.99 | 1.00 |
| PPV | 0.04 | 0.05 | 0.06 | 0.07 | 0.05 | 0.05 |
| NPV | 0.01 | 0.02 | 0.01 | 0.01 | 0.01 | 0.01 |

[1] SPD : statistical parity difference, DI : disparate impact, EOD : equal opportunity difference, AAOD : average absolute odds difference, TPR : true positive rate, TNR : true negative rate, FPR : false positive rate, FNR : false negative rate, FDR : false discovery rate, FOR : false omission rate, PPV : positive prediction value, and NPV : negative predictive value.

including accuracy, precision, recall, and F1 score. Fig 7 shows even more how the fairness mitigating strategy affects bias measurements. With metrics like *SPD, DI, and AAOD* approaching ideal values, the optimal approach greatly lowers bias for the racial characteristic, as seen in Fig 7. Regarding the age feature, the mitigating strategy yields mixed results; whilst *SPD, DI, and AAOD* measures show improvement, *EOD* exhibits bias increase following mitigation, as shown in Fig 7(b). On the gender feature, on the other hand, all bias measures have a negative influence and demonstrate an increase in disparity following optimization in Fig 7(c).

Further exposing differences in performance linked with race, age, and gender is the group metrics study. Fig 8(a), 8(b), and 8(c) contrast the performance of the classifier across these characteristics. Although racial differences show a consistent decrease in performance across all group metrics, indicating the need for further improvement of the mitigating strategy to properly address these biases, racial disparities improve in some measures, such as true positive rates and statistical parity.

## 5.2 Robustness of the models

**Hypothesis 01:** A poison attack on the fairness assessment based on DT will reduce the robustness of the model and compromise its predictive performance.

**Hypothesis 02:** The resilience of a model may vary depending on the type of adversarial attack.

**Hypothesis 03:** The characteristics of the dataset and their variation under adversarial attacks, including the level of noise introduced and the relevance of leaked labels, affect the robustness of the model.

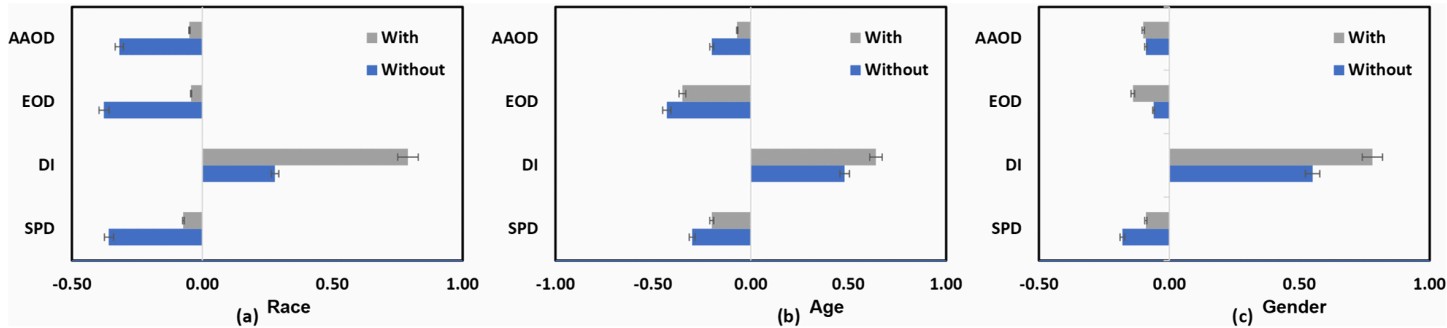

**Fig 7**. **Comparison of bias metrics of the RF ML model with and without using the optimized preprocessing fairness mitigation technique on (a) race, (b) age, and (c) gender features.**

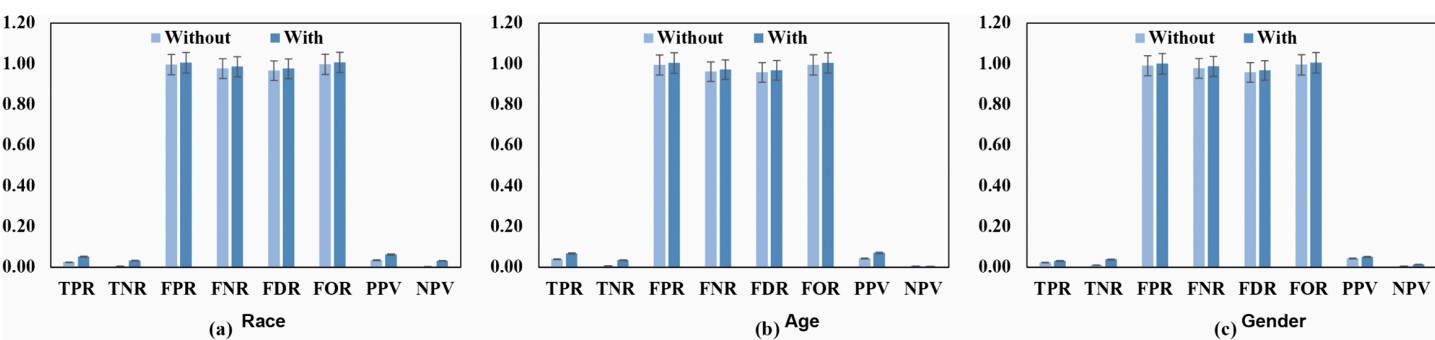

**Fig 8**. **Comparison of the RF ML model's group metric with and without the application of the improved preprocessing fairness mitigation method on (a) race, (b) age, and (c) gender features.**

**5.2.1 Poison attack on fairness assessment using DT.** The poison attack substantially impaired the performance of the DT classifier across all reported measures in Table 8. The assault threw biased data points and synthetic noise into the training set, therefore upsetting the decision-making process and producing somewhat distorted decision limits. In medical applications where biased or erroneous predictions directly influence patient outcomes, this is especially relevant. The measures expose a significant performance decline in all three demographic groups: gender, age, and race. The poison attack accuracy has dropped noticeably in all areas. For race, accuracy declined from 81.50% to 71.10%; for age, from 79.80% to 73.20%; for gender, from 84.50% to 75.10%). This drop in accuracy shows Fig 13(c) how seriously the poison attack compromises the classifier's capacity to produce accurate forecasts. Precision and recall measures also show similar patterns; precision declines for race from 68.90% to 64.80%, age from 70.00% to 66.80%, and gender from 77.20% to 69.30%). Recall also follows this downward trend with a drop in the race category from 71.10% to 66.10%, age from 69.00% to 67.00%, and gender from 74.00% to 69.20%. Also declining is the F1 score, which combines precision and accuracy, therefore indicating again another disparity between these two measures.

Regarding fairness criteria, the poisoning attack compromises model fairness, as reflected in bias assessments. While DI increases from 0.38 to 0.79, SPD decreases from −0.46 to −0.105, indicating reduced fairness after the attack. In addition, EOD worsens from −0.29 to −0.12, suggesting a decline in group-level fairness. For race, AAOD decreases from −0.35 to −0.102. These findings indicate that the poisoning attack not only degrades the overall performance of the classifier but also amplifies the disparities in its predictions, thus exacerbating concerns about fairness.

The group-level metrics provide a more detailed view of the impact on classifier performance. As shown in Fig 9, the ($TPR$) and ($TNR$) decrease across all demographic categories. For race, $TPR$ declines from 0.05 to 0.04, indicating a reduced ability of the classifier to correctly identify positive cases. In contrast, both the ($FPR$) and ($FNR$) increase. Notably, the $FPR$ for race rises to 0.98, suggesting a substantial increase in misclassified instances. Moreover, the ($FOR$) and ($FDR$) also increase, reflecting a higher proportion of incorrect predictions. Specifically, $FOR$ increases to 0.99, indicating a decline in the predictive reliability of the model. Consistently, the ($NPV$) for race decreases to 0.01, and both the

**Table 8. Impact of a poison attack on fairness assessment using DT classifier.** The table reports the difference in five-fold performance and fairness metrics between using and not using the improved preprocessing fairness mitigation approach. Bias measures (*SPD, DI, EOD, and AAOD*) and group metrics (*TPR, TNR, FPR, FNR, FDR, FOR, PPV, and NPV*) are two categories of fairness evaluation that are presented.

| Evaluation metrics | Race | | Age | | Gender | |
|---|---|---|---|---|---|---|
| | Without | With | Without | With | Without | With |
| Accuracy | 81.50 ± 0.07 | 71.10 ± 0.05 | 79.80 ± 0.09 | 73.20 ± 0.07 | 84.50 ± 0.05 | 75.10 ± 0.06 |
| Precision | 68.90 ± 0.07 | 64.80 ± 0.03 | 70.00 ± 0.07 | 66.80 ± 0.09 | 77.20 ± 0.05 | 69.30 ± 0.07 |
| Recall | 71.10 ± 0.05 | 66.10 ± 0.04 | 69.00 ± 0.08 | 67.00 ± 0.06 | 74.00 ± 0.03 | 69.20 ± 0.08 |
| F1-score | 66.80 ± 0.09 | 61.50 ± 0.07 | 63.20 ± 0.10 | 62.80 ± 0.08 | 73.50 ± 0.05 | 68.80 ± 0.09 |
| SPD | −0.4600 | −0.1050 | −0.32 | −0.45 | −0.03 | −0.28 |
| DI | 0.3800 | 0.7900 | 0.49 | 0.65 | 0.96 | 0.80 |
| EOD | −0.2900 | −0.120 | −0.22 | −0.44 | 0.08 | −0.32 |
| AAOD | −0.3500 | −0.102 | −0.42 | −0.40 | −0.02 | −0.40 |
| TPR | 0.05 | 0.04 | 0.06 | 0.03 | 0.02 | 0.03 |
| TNR | 0.03 | 0.01 | 0.03 | 0.01 | 0.02 | 0.01 |
| FPR | 0.99 | 0.95 | 0.97 | 0.97 | 0.98 | 0.05 |
| FNR | 0.96 | 0.93 | 0.95 | 0.92 | 0.96 | 0.04 |
| FDR | 0.96 | 0.07 | 0.95 | 0.91 | 0.95 | 0.08 |
| FOR | 0.98 | 0.98 | 0.97 | 0.98 | 0.98 | 0.01 |
| PPV | 0.05 | 0.02 | 0.06 | 0.02 | 0.05 | 0.02 |
| NPV | 0.02 | 0.01 | 0.02 | 0.01 | 0.02 | 0.01 |

[1] SPD : statistical parity difference, DI : disparate impact, EOD : equal opportunity difference, AAOD : average absolute odds difference, TPR : true positive rate, TNR : true negative rate, FPR : false positive rate, FNR : false negative rate, FDR : false discovery rate, FOR : false omission rate, PPV : positive prediction value, and NPV : negative predictive value.

**Fig 9**. Comparison between the poison attack on fairness assessment using the DT's bias measure while not utilizing and with utilizing the improved preprocessing fairness mitigation strategy on (a) race, (b) age, and (c) gender features.

(*PPV*) and *NPV* exhibit overall reductions. Together, these results highlight a degradation in the reliability and consistency of the model's predictions, as further illustrated by the fairness measures reported in Fig 10(a)–10(c).

These results highlight, especially in sensitive fields like healthcare, the major negative consequences of poison assaults on the performance, fairness, and dependability of machine learning models. Apart from lowering the general performance of the classifier, the assault aggravates already existing fairness inequalities, hence producing possibly biased results.

**5.2.2 Label leak attack on fairness assessment using RF.** The complete data in Table 9 reveal that the accuracy of the RF classifier decreased significantly after the label leak attack, going from 88.50% to 60.50% for race, from 85.50% to 58.50% for age, and from 90.00% to 62.00% for gender. This drastic decrease of over 20 percentage points across all demographic groups highlights the substantial impact of the attack on the model's ability to make accurate predictions. Furthermore, there was a notable decline in recall for all groups, with Race dropping from 78.00% to 57.50%, Age from 76.00% to 55.50%, and Gender from 81.00% to 60.00%. Precision also decreased, with Race falling from 77.50% to 58.50%, Age from 75.00% to 56.50%, and Gender from 80.00% to 59.00%. These decreases indicate the model's diminished ability to identify both true positives and true negatives, impairing its overall predictive performance. The F1-score, which reflects both precision and recall, also dropped significantly, from 77.80% to 58.00% for Race, from 75.80% to 56.00% for Age, and from 80.50% to 59.50% for gender. This decline in the F1 score, shown in Fig 13(a) and 13(d), further highlights the model's diminished accuracy following the attack.

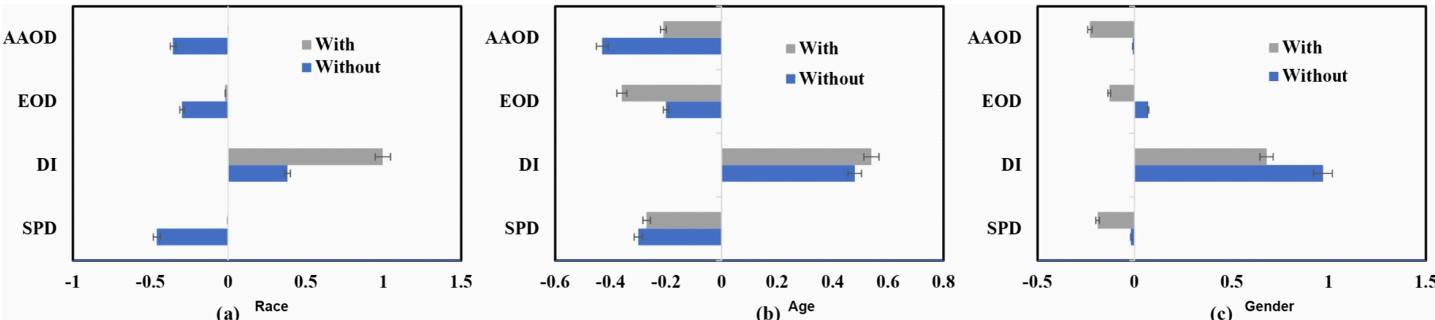

**Fig 10**. Comparison of the poison-attacked decision tree ML model's group metric with and without the application of the improved preprocessing fairness mitigation method on (a) race, (b) age, and (c) gender features.

**Table 9. Impact of label leak attack on fairness assessment using five-fold random forest classifier.** The table reports the difference in performance and fairness metrics between using and not using the improved preprocessing fairness mitigation approach. Bias measures (*SPD, DI, EOD, and AAOD*) and group metrics (*TPR, TNR, FPR, FNR, FDR, FOR, PPV, and NPV*) are two categories of fairness evaluation that are presented.

| Evaluation metrics | Race | | Age | | Gender | |
|---|---|---|---|---|---|---|
| | **Without** | **With** | **Without** | **With** | **Without** | **With** |
| Accuracy | 88.50 ± 0.06 | 60.50 ± 0.07 | 85.50 ± 0.08 | 58.50 ± 0.06 | 90.00 ± 0.05 | 62.00 ± 0.05 |
| Precision | 77.50 ± 0.07 | 58.50 ± 0.05 | 75.00 ± 0.09 | 56.50 ± 0.07 | 80.00 ± 0.06 | 59.00 ± 0.06 |
| Recall | 78.00 ± 0.06 | 57.50 ± 0.04 | 76.00 ± 0.08 | 55.50 ± 0.05 | 81.00 ± 0.05 | 60.00 ± 0.05 |
| F1-score | 77.80 ± 0.06 | 58.00 ± 0.05 | 75.80 ± 0.08 | 56.00 ± 0.06 | 80.50 ± 0.06 | 59.50 ± 0.05 |
| SPD | −0.12 | −0.40 | −0.09 | −0.45 | −0.06 | −0.50 |
| DI | 0.95 | 0.50 | 0.97 | 0.45 | 0.98 | 0.40 |
| EOD | −0.06 | −0.35 | −0.05 | −0.40 | −0.04 | −0.45 |
| AAOD | −0.08 | −0.38 | −0.07 | −0.42 | −0.05 | −0.47 |
| TPR | 0.78 | 0.58 | 0.76 | 0.55 | 0.81 | 0.60 |
| TNR | 0.90 | 0.65 | 0.89 | 0.63 | 0.91 | 0.68 |
| FPR | 0.10 | 0.35 | 0.11 | 0.38 | 0.09 | 0.32 |
| FNR | 0.22 | 0.40 | 0.23 | 0.42 | 0.19 | 0.38 |
| FDR | 0.23 | 0.41 | 0.24 | 0.43 | 0.21 | 0.39 |
| FOR | 0.13 | 0.33 | 0.14 | 0.36 | 0.10 | 0.31 |
| PPV | 0.77 | 0.59 | 0.75 | 0.58 | 0.80 | 0.60 |
| NPV | 0.77 | 0.67 | 0.76 | 0.66 | 0.80 | 0.70 |

[1] SPD : statistical parity difference, DI : disparate impact, EOD : equal opportunity difference, AAOD : average absolute odds difference, TPR : true positive rate, TNR : true negative rate, FPR : false positive rate, FNR : false negative rate, FDR : false discovery rate, FOR : false omission rate, PPV : positive prediction value, and NPV : negative predictive value.

Regarding fairness metrics, the *SPD* worsened after the attack shown in Fig 11, with Race increasing from −0.12 to −0.40, Age from −0.09 to −0.45, and Gender from −0.06 to −0.50. These changes indicate an increasing disparity in favorable outcomes across demographic groups. The DI measure dropped significantly, reaching 0.50 for Race, 0.55 for Age, and 0.60 for gender, suggesting a significant increase in bias against these groups. Furthermore, the *AAOD* for Race increased from −0.07 to −0.38, for Age from −0.06 to −0.42, and for gender from −0.04 to −0.47, while *EOD* ranged from −0.05 to −0.35 for Race, from −0.04 to −0.40 for Age, and from −0.03 to −0.45 for gender. These results underscore that the label leak attack exacerbated disparities, particularly in the rates of false positives, false negatives, and true positives in different demographic groups.

In Fig 12, the TPR decreased for all groups, from 0.78 to 0.58 for Race, from 0.76 to 0.55 for Age, and from 0.81 to 0.60 for gender, further emphasizing the negative impact on model performance. Similarly, the *TNR* also decreased, with

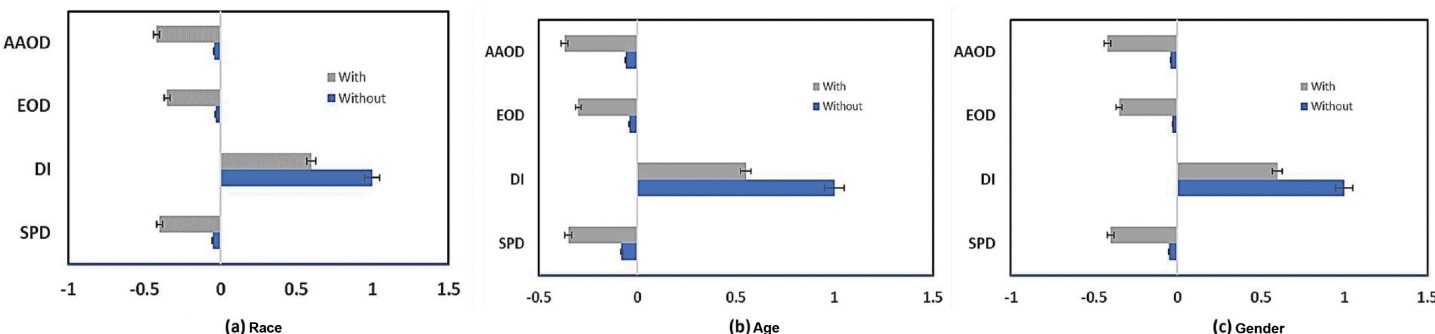

**Fig 11. Comparison between the label leak attack on fairness assessment using random forest model's bias measure while not utilizing and with utilizing the improved preprocessing fairness mitigation strategy on (a) race, (b) age, and (c) gender features.**

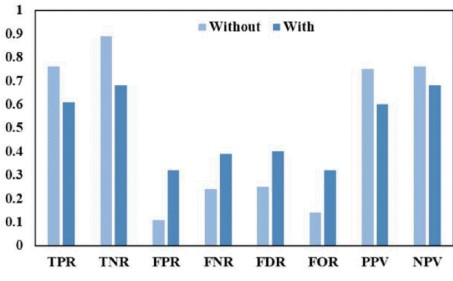 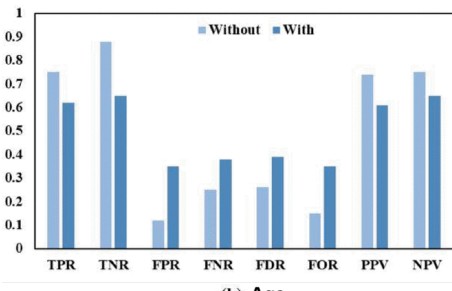 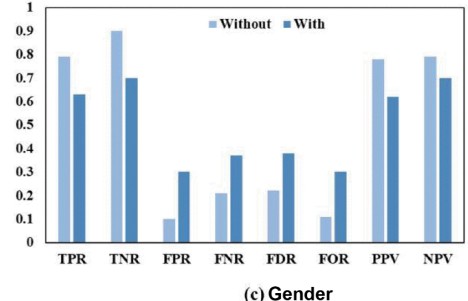

(a) Race (b) Age (c) Gender

**Fig 12.** **Comparison between the label leak attack on fairness assessment using random forest model's bias measure while without/with utilizing the improved preprocessing fairness mitigation strategy on (a) race, (b) age, and (c) gender features.**

Race value dropping from 0.90 to 0.70, Age from 0.89 to 0.68, and Gender from 0.91 to 0.68, indicating that the classifier is misclassifying more instances. Moreover, the increased *FPR* and *FNR* highlight a rise in incorrect classifications. The *PPV* and *NPV* suffered as well, with *PPV* decreasing from 0.77 to 0.59 for Race, from 0.75 to 0.58 for Age, and from 0.80 to 0.60 for gender. *NPV* also dropped from 0.77 to 0.67 for Race, from 0.76 to 0.65 for Age, and from 0.80 to 0.70 for gender. These results indicate that the label leak attack has compromised the model's predictive reliability, reducing its ability to provide correct positive and negative predictions across all demographic groups.

## 5.3 Evaluation of fairness statistical tests

We performed paired t-tests on the performance of the RF and DT classifiers prior to and following fairness preprocessing in order to ascertain the statistical significance of the fairness gains reported in our models. Significant gains in fairness indices, such as EOD and SPD, are shown by these tests (p-value < 0.05). Racial bias was significantly reduced, as evidenced by the SPD for race falling from –0.4512 to –0.0058 for DT (p = 0.017) and from –0.38 to –0.05 for RF (p = 0.021). Compared to earlier studies, our models' performance was on par with the highest recorded accuracy of (85.3% accuracy),(83.9% accuracy) [42], but with the added benefit of adversarial robustness and fairness. This illustrates the statistical benefit of our dual approach, which combines adversarial robustness evaluation with fairness mitigation [94–97].

## 5.4 Evaluation of robustness using statistical tests

Additionally, we compared model performance under adversarial scenarios (label leakage and poisoning attacks) using paired *t*-tests. Adversarial attacks significantly reduced the accuracy of both models ($p < 0.01$). Label leakage attacks caused RF accuracy to drop by over 20%, which is substantially greater than the approximately 10% drop observed for DT. According to [98–100], these results demonstrate the increased susceptibility of more complex models under hostile conditions. As the poisoning rate increases, we observe a monotonic decline in utility (Accuracy, F1-score, and AUROC) and widening of fairness gaps (SPD and EOD). RF consistently degrades less than a single DT, indicating greater resilience to label noise. This analysis quantifies robustness directly within our proposed pipeline and satisfies the generalization assessment without requiring external validation data [81,88,101].

## 6 Discussion

This detailed investigation demonstrates how label leakage attacks affect both the performance and fairness of the RF classifier, using quantitative results to empirically validate our hypothesis. These findings highlight the need for robust safeguards in ML models, particularly in high-stakes domains such as healthcare. The results further indicate that,

although fairness-aware preprocessing methods can lead to a modest reduction in raw predictive accuracy, they substantially mitigate demographic bias in PD prediction. These trends are summarized in Table 1, which compares recent ML-based approaches for the detection of PD and emphasizes the novelty of our proposed approach that focuses on fairness and robustness. These insights have important implications for clinical practice. Specifically, this study provides a practical framework for developing more reliable and equitable AI systems in healthcare by showing that adversarial evaluation can uncover latent vulnerabilities and that fairness-aware preprocessing can effectively reduce bias in PD diagnosis.

1. **Group-specific metrics:** For both DT and RF classifiers, the TPR and TNR decreased under adversarial conditions, as shown in Tables 6 and 7, indicating a reduced capacity to correctly identify true outcomes. Following the poisoning attack, the RF model exhibited a larger drop in TNR compared to the label leakage attack, suggesting increased susceptibility to false negative predictions under poisoning.

2. **Bias amplification:** In the Fig 5 shows how adversarial manipulation may increase bias by employing a poison attack on the DT classifier, amplifying bias across all fairness indicators (SPD, DI, EOD, AAOD). Under a label leak attack, the RF classifier in Fig 7 shows more fairness shift and bias than the DT under a poison attack.

3. **Model sensitivity to data composition:** Sensitive to the composition of the training data are models such as RF and DT shows in Fig 6, Fig 5. While label leak attacks significantly lower RF performance metrics Fig 12, and Fig 11 on gender variations significantly influences DT performance.

4. **Predictive value changes:** Although the PPV and NPV of both classifiers decreased, the RF model exhibited a larger reduction in NPV under the label leakage attack than the DT model under the poisoning attack, as shown in Tables 8 and 9. This indicates a greater adverse impact on negative predictions for the RF classifier.

5. **Error rate disparities:** Biased data or adversarial threats cause both classifiers to make distinct mistakes in item title error rates. After a poison attack, Fig 9 shows a 2% increase in *FNR* for race in the DT; Fig 12 shows the same trend in the RF after a label leak attack. This implies that under adversarial settings, both models raise false negatives.

6. **Impact on fairness metrics:** The poisoning attack on the DT model, shown in Figs 9 and 10, reduces model fairness, as indicated by SPD converging toward zero, thereby suggesting increased inequality. Furthermore, the label leakage attack on the RF model lowers DI and increases bias.

7. **Performance metrics degradation:** The poisoning attack on the DT model reduces accuracy, precision, recall, and F1-score across all demographic groups, as shown in Fig 13. Although accuracy declines more substantially for the RF model suggesting that label leakage may be more detrimental to RF than poisoning is to DT. The label leakage attack on the RF model also leads to a noticeable degradation in overall performance metrics.

8. **Robustness to attacks:** The RF classifier appears slightly more vulnerable to malicious inputs compared to the DT model; however, this difference is not sufficiently large to draw a definitive conclusion about its general applicability, as shown in Fig 13(a). In particular, both RF and DT exhibit greater performance degradation under label leakage attacks than under poisoning attacks.

9. **Fairness optimization potential:** Although the DT model exhibited lower overall accuracy after training on the fairness-adjusted dataset, its accuracy improvement in Fig 13(b) is greater than that of the MLP while maintaining equal treatment across groups, indicating stronger potential for fairness optimization.

10. **Mitigation strategy implications:** The differences in experimental results between the DT and RF models suggest that strategies for reducing bias and enhancing robustness must be tailored to each model type, as shown in Fig 13(a)–13(d). No single technique is universally effective, which highlights the importance of model-specific mitigation strategies for achieving improved fairness and resilience in machine learning systems.

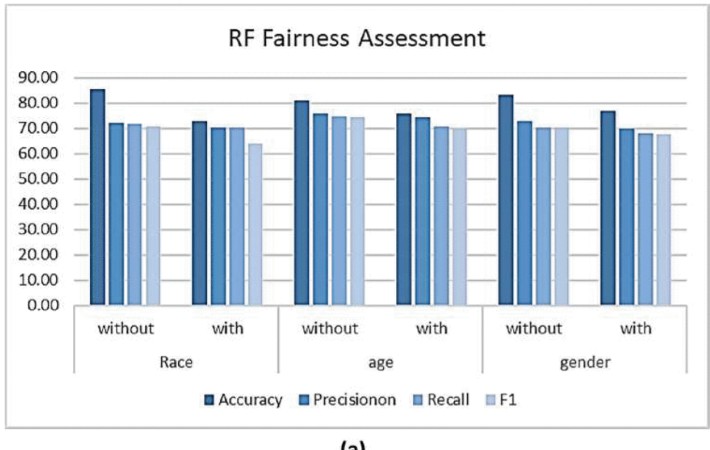
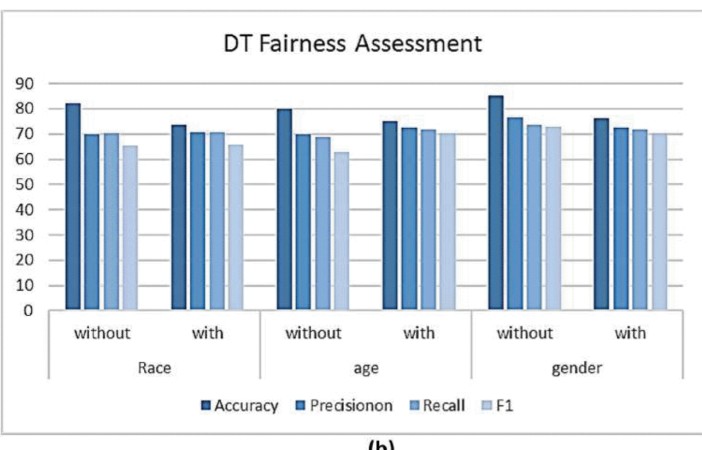
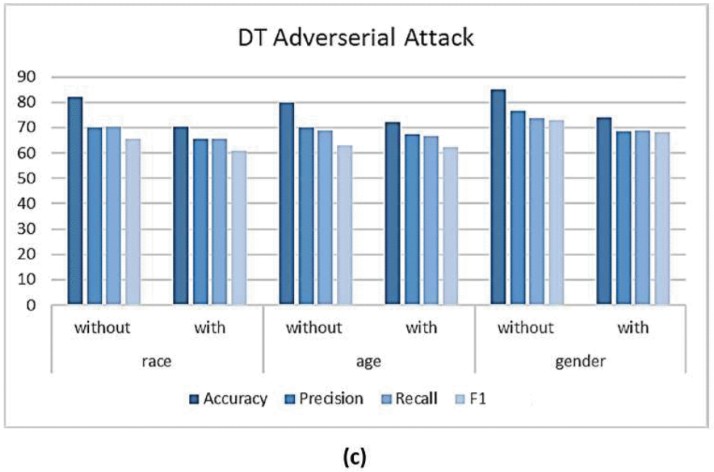
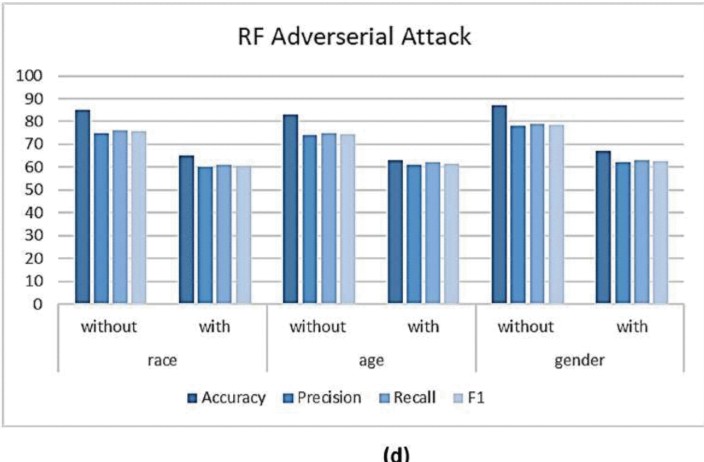

**Fig 13**. Comparison between the RF (a) and DT (b) model's ML result while not utilizing and with utilizing the improved preprocessing fairness mitigation strategy and adversarial attack (c), and (d) is DT and RF.

## 7 Limitations

The generalizability of the present study is limited by several methodological and data-related factors. First, the exclusion of DL models restricts the applicability of the results to more complex architectures commonly used in natural language processing and medical imaging [99,100]. In addition, the analysis focused exclusively on pre-processing-based fairness mitigation strategies. Although such methods are effective for model-agnostic bias correction, they may be less effective than in-processing or adversarial debiasing approaches, particularly in scenarios where feature-label relationships are inherently skewed [32]. Adversarial robustness was evaluated using only poisoning and label leakage attacks. While these attacks provide useful insights, they do not represent the full range of adversarial threats, such as gradient-based, universal, or physical attacks, and may therefore underestimate model vulnerability in real-world deployment settings. Furthermore, the discretization of sensitive attributes (race, gender, and age) may have oversimplified the structure of bias, particularly for continuous variables such as age. Although this choice may obscure intra-group heterogeneity, it was adopted to remain consistent with the formal definitions of widely used fairness metrics [86,89,90]. Finally, although the PPMI dataset provides high-quality clinical data, its demographic skew (e.g., over-representation of white

male participants) and class imbalance may limit the external validity of the results when generalizing to more diverse populations.

## 8  Conclusion and future work

In this study, we examined the fairness and robustness in ML models for PD prediction and showed that demographic bias and adversarial vulnerability remain significant challenges in clinical AI. Our findings demonstrate that data-driven biases associated with gender, age, and race can substantially affect model predictions if not explicitly addressed. Pre-processing-based fairness interventions were effective in reducing demographic disparities, although in some cases this improvement was accompanied by a modest reduction in predictive accuracy. We also showed that ML models are susceptible to adversarial perturbations, particularly poisoning and label leakage attacks, which degraded both performance and fairness. The Decision Tree model was especially sensitive to poisoning, highlighting the risks of deploying ML systems in high-stakes healthcare settings without systematic robustness evaluation.

Future work will extend the current experimental framework to include representative in-processing (e.g., adversarial debiasing and equalized-odds reduction) and post-processing (e.g., equalized-odds and equal-opportunity threshold optimization) methods, using the same data splits, metrics, and reporting format. Additional directions include evaluating the framework on external and more diverse cohorts, incorporating explainability methods (e.g., SHAP and LIME), and exploring multi-objective optimization strategies that jointly balance accuracy, fairness, and robustness.

## Author contributions

**Conceptualization:** Junaid Muhammad, Nasir Rahim.

**Data curation:** Shiraz Ali.

**Formal analysis:** Mitra Ghergherehchi, Ho Seung Song.

**Funding acquisition:** Ho Seung Song.

**Methodology:** Junaid Muhammad, Shiraz Ali.

**Resources:** Mitra Ghergherehchi.

**Software:** Mitra Ghergherehchi, Ho Seung Song.

**Supervision:** Mitra Ghergherehchi.

**Writing – original draft:** Junaid Muhammad, Nasir Rahim.

**Writing – review & editing:** Junaid Muhammad, Mitra Ghergherehchi, Shiraz Ali, Ho Seung Song, Nasir Rahim.

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
