## [Decision Letter · Decision Letter 0]

23 Apr 2025

PONE-D-25-14929

Trustworthy AI for medical decisions: Adversarial robust and fair machine learning prediction for Parkinson’s disease

PLOS ONE

Dear Dr. Rahim,

Thank you for submitting your manuscript to PLOS ONE. After careful consideration, we feel that it has merit but does not fully meet PLOS ONE’s publication criteria as it currently stands. Therefore, we invite you to submit a revised version of the manuscript that addresses the points raised during the review process.

**ACADEMIC EDITOR:**

Required:

The abstract should more clearly state the study’s objectives, dataset characteristics, and the rationale for including adversarial attack analysis.In Section 3.2.2, several claims—particularly regarding model robustness, federated learning, and explainability—require proper justification and supporting references.The discussion on bias related to gender, age, and race should be more clearly substantiated with examples and explanations.The mention of "research focused on algorithmic fairness" in Section 3.3 lacks citation, and Section 3.4.2 appears misaligned with its intended focus on group fairness.A clearer distinction between sections 4.2 and 4.4 is needed as they currently overlap in content.Furthermore, the limitations of the experimental setup should be explicitly addressed.Key related works and comparisons, particularly those using similar datasets and standard models like SVMs or neural networks, should be included.Feature selection choices—especially the exclusion of age and race—should be justified.Metric inconsistencies should be addressed, with valid ranges and statistical significance testing included.Imbalances in class distribution should be noted and mitigated appropriately.Data availability should align with reproducibility standards, and the ethical treatment of sensitive attributes—such as discretization of age or race—requires clear explanation.It is important to explicitly note limitations, such as the exclusion of deep learning models, and discuss strategies to manage accuracy-fairness trade-offs in future work.Language should be polished for grammatical accuracy and reduced complexity.Formatting and consistency in references, notation, and tables should also be reviewed

Recommendations:

The paper would benefit from increased originality and refinement in both content and presentation.The authors are encouraged to compare their findings with current state-of-the-art methodologies to better contextualize their contribution.The introduction contains an extensive list of contributions, which could be narrowed to emphasize the main innovations.The concept of fairness is frequently mentioned but not always supported by concrete evidence.A comparative table summarizing models, datasets, pros, cons, and performance would be a valuable addition.Certain sections could be restructured for clarity.Also, the methodology and results sections would benefit from added detail on experimental setup, parameters, and statistical significance.A dedicated discussion section analyzing the results would strengthen the manuscript.Methodological transparency can be improved by detailing fairness mitigation algorithms, feature selection techniques, and adversarial attack parameters.

We look forward to receiving your revised manuscript.

Kind regards,

Rizik M. H. Al-Sayyed, Ph.D.

Academic Editor

PLOS ONE

“This work was supported by the National Research Foundation of Korea (NRF) grant 716

funded by the Korea government (MSIT) (No. RS-2023-00221186) and also supported 717

by the National Research Foundation of Korea (NRF) grant funded by the Korea 718

government (MSIT) (No. RS-2022-00166733).”

Additional Editor Comments:

Thank you for your submission. While your work raising one of the important issues in healthcare AI, especially pertaining to fairness and adversarial robustness, requires substantial work. The reviewers determined that the manuscript fails to be coherent, methodologically sophisticated, and scientifically thorough. Primary issues include lack of reasonable justification for certain arguments (especially in Sections 3.2.2, 3.3), not addressing contemporary relevant literature, insufficient detailing of bias, redundancy in data pre-processing explanations, and no benchmarking against contemporary literature. Add the problem, dataset, and technologies used in the abstract. Furthermore, inconsistencies such as having FPR greater than one, vague defining of terms such as mitigation of fairness, adversarial attacks, or mistreatment as well as fuzzy explanations should be fixed. Revise the document so that all the statements made are supported by relevant academic literature. Make sure claims are backed by references, redefine the results and discussions as well as limit the conclusions to frameworks set out in the objectives. More focusing on aspect of providing rationale for why concepts such as mitigation requires more elaboration. The manuscript requires, at minimum, this level of overhaul in conjunction with enhancing the scientific as well as editorial quality of the manuscript.

Reviewers' comments:

Reviewer's Responses to Questions

**Comments to the Author**

1. Is the manuscript technically sound, and do the data support the conclusions?

Reviewer #1: No

Reviewer #2: Yes

Reviewer #3: Partly

2. Has the statistical analysis been performed appropriately and rigorously?

Reviewer #1: Yes

Reviewer #2: No

Reviewer #3: Yes

3. Have the authors made all data underlying the findings in their manuscript fully available?

Reviewer #1: Yes

Reviewer #2: No

Reviewer #3: No

4. Is the manuscript presented in an intelligible fashion and written in standard English?

Reviewer #1: No

Reviewer #2: Yes

Reviewer #3: No

5. Review Comments to the Author

Reviewer #1: 1. Your work lacks sufficient originality and does not meet the minimum quality standards of PLOS ONE Journal.

2. You need to enhance the English writing in all paper sections. Several statements and paragraphs are not understood.

3. Rewrite the abstract section to specify the problem directly and describe your solution in brief.

4. Compare your results with state-of-the-art approaches.

5. In Adversarial Robustness section 3.2.2, you draw conclusions without justifications. (There is a lot of bias in the information when it comes to gender, age, and race, which shows how these traits are uneven across classes. Compared to gender features, age and race features are more skewed). Explain how bias exists?

6. In section 3.2.2 there are conclusions without justification, such as:

• Adversarial training is one of the strong training methods that boost model generalization and manipulation resistance.

• Resilience added to federated learning systems increases data privacy and model efficiency.

• Explainable artificial intelligence (XAI) may highlight the evolution and upkeep of robustness.

• Furthermore, standardizing model robustness assessment will be a new criterion for healthcare artificial intelligence.

7. In section 3.3, you wrote (In research focused on algorithmic fairness, which involves the removal of biases as illustrated in Figure 1) which research do you mean? No reference is given.

8. The content of section 3.4.2 Group fairness does not relate to Group fairness, it relates to the model’s assessment.

9. Section 4.2 Data prepossessing and section 4.4 Data preparation have the same contents. What are the differences?

10. Experiments limitations should be discussed.

Reviewer #2: This study presented a CAD for Parkinson’s disease that studied the impact on health-related quality of life and revealed disparities based on gender, age, and race. The authors applied Random Forest (RF) and Decision Tree (DT) on the Parkinson’s disease dataset.

There are some comments that need to be considered in the article:

The abstract lacks clarity regarding the objective of using machine learning techniques. Additionally, a description of the dataset is also not given in the abstract. Adversarial attacks are not clear, and why to add it to the data.

In the introduction,

The number of primary contributions mentioned is huge; it is better to be reduced to your main contribution only.

The authors utilize the word fairness many times in many sections without proving if they fairly use it. For example, in the literature review they make, however, there are many articles not included in the study.

in subsection 2.1, "such as those covered here," unclear” pronoun those”

In Related Works, it is better to add the most related models that have already been applied to the same dataset of Parkinson. Did any authors utilize other standard ML techniques such as SVM or neural networks?

It is better to add a table with the ML techniques, dataset, pros and cons of each study, and performance.

Why add this "section 3.3 Mitigate ML bias"?

In methodology, just mention the main methods that you are utilizing in your study or that you wish to compare with only.

3.2.1 Model robustness: In lines 191–192, it is better to add a reference for your claim.

Add reference for lines 378 and 379: XGBoost and SULOV. Which one did you utilize in your study?

In data splitting, there is no reason to make 5-fold cross-validation after the initial separation of the data.

In the data collection, the number of features is around 15. Could you please explain the reasoning behind using feature selection?

And if you used FS, why did you exclude age and race? You can let FS method selects the most valuable features.

In the results section,

First, it is better to add in the experiment setup.

The parameters for the decision tree and random forest

Comparing your results with the most recent studies already discussed in your related works is preferable.

and it is better to compute the significance of your obtained results.

In addition, it is better to add a discussion section, which discusses the finding behind the obtained results.

Reviewer #3: This manuscript significantly contributes to the discourse on trustworthy AI in healthcare, and by addressing the following outlined points, the authors can enhance the rigor, clarity, and impact of their work. The study’s focus on fairness and adversarial robustness is commendable, and the suggested revisions will help it meet the high standards expected by PLOS ONE. Therefore, it is recommended for acceptance with minor revisions and a medium confidence level.

Minor Revisions

1. Statistical Analysis and Metric Validity:

o Address Metric Anomalies: Several tables report implausible values (e.g., FPR >1). Revise calculations and ensure all metrics (FPR, FNR) are within valid ranges (0–1).

o Statistical Significance Testing: To confirm significance, include p-values or confidence intervals for key results (e.g., accuracy drops post-mitigation/adversarial attacks).

o Class Imbalance: Acknowledge and mitigate the impact of the PD: HC imbalance (≈3:2) using techniques like SMOTE, stratified sampling, or synthetic data generation.

2. Methodological Clarity:

o Fairness Mitigation Pipeline: Provide explicit details on the "optimized preprocessing fairness mitigation" algorithm (e.g., reweighting, adversarial debiasing). Include hyperparameters and code references if available.

o Feature Selection: Clarify the implementation of SULOV and recursive XGBoost (e.g., library versions, selection criteria).

o Adversarial Attacks: Specify perturbation magnitudes (e.g., noise levels in poison attacks) and success rates (e.g., % adversarial examples generated).

3. Data Availability:

o Remove Ambiguity: Delete the redundant "available on request" statement unless legally/ethically justified.

o Share Processed Data: Deposit preprocessed/feature-engineered datasets in a public repository (e.g., Zenodo) to ensure reproducibility.

4. Ethical and Bias Considerations:

o Sensitive Attribute Binarization: Justify thresholds for discretizing age, race, and gender. For example, clarify how age groups were defined (e.g., <60 vs. ≥60).

o Causal Mechanisms: Discuss how adversarial attacks amplify bias (e.g., label leaks correlating with SPD decline) rather than solely reporting correlations.

5. Language and Grammar:

o Correct grammatical errors (e.g., "data was" → "data were," "prepossessing" → "preprocessing").

o Simplify jargon-heavy sections (e.g., "quantum adversarial machine learning") with brief definitions or examples.

o Reduce redundancy (e.g., repetitive mentions of "fairness-accuracy trade-offs" in the Discussion).

6. Formatting Consistency:

o Ensure uniform citation style (e.g., "[13]" vs "as shown in [13]").

o Standardize mathematical notation (e.g., italics for variables: FPR vs. FPR).

7. Result Presentation:

o Tables 3–6: Reformat to avoid split tables across pages and ensure alignment.

o Figures: Improve resolution and label clarity (e.g., axis labels in Figures 5–8).

8. Discussion Limitations:

o Explicitly state the exclusion of deep learning models as a limitation and its impact on generalizability.

o Discuss potential solutions to accuracy-fairness trade-offs (e.g., hybrid models, post-hoc calibration).

6. PLOS authors have the option to publish the peer review history of their article (what does this mean?). If published, this will include your full peer review and any attached files.

Reviewer #1: No

Reviewer #2: **Yes:** Dheeb Albashish

Reviewer #3: No

---

## [Author Response · Author response to Decision Letter 1]

11 Jun 2025

Response to the Reviewers on “Trustworthy AI for medical decisions: Adversarial robust and fair machine learning prediction for Parkinson’s disease”

Junaid Muhammad, Mitra Ghergherehchi, Shiraz Ali, Ho Seung Song, Nasir Rahim

Dear editor,

We thank you and the reviewers for your positive comments that helped us a lot to enrich the overall quality of this study. We have considered the reviewer’s comments carefully and tried our best to incorporate all the comments/suggestions to demonstrate the effectiveness of our proposal.

To address each comment carefully, we would like to respond to them point by point. Please note that all updates in the manuscript have been highlighted in red color.

Kind regards,

Authors

Reviewer’s comment

The abstract should more clearly state the study’s objectives, dataset characteristics, and the rationale for including adversarial attack analysis.

Author’s response

Thank you for your valuable comment.

We have revised the abstract to more clearly state the study’s objectives, describe the PPMI dataset characteristics, and explain the rationale for incorporating adversarial attack analysis. This change enhances the clarity and purpose of our work from the outset.

Reviewer’s comment

In Section 3.2.2, several claims—particularly regarding model robustness, federated learning, and explainability—require proper justification and supporting references.

Author’s response

Thank you.

In Section 3.2.2, we have revised the claim regarding model robustness, federated learning, and explainability by providing proper justification and supporting references from recent literature [98-105]. These additions ensure that the statements are well-supported and grounded in current research, improving the clarity and credibility of the section.

Reviewer’s comment

The discussion on bias related to gender, age, and race should be more clearly substantiated with examples and explanations.

Author’s response

Thank you.

We have updated the discussion on data bias related to gender, age, and race in Section 3.2.2 to provide more concrete examples and detailed explanations. This revision ensures that the impact of bias is better substantiated and aligns with the study's objectives.

Reviewer’s comment

The mention of "research focused on algorithmic fairness" in Section 3.3 lacks citation, and Section 3.4.2 appears misaligned with its intended focus on group fairness.

Author’s response

Thank you.

We have updated Section 3.3 and included relevant citations supporting the discussion on algorithmic fairness. Additionally, we have revised Section 3.4.2 to better align with its intended focus on group fairness, ensuring clearer structure and consistency in the manuscript.

Reviewer’s comment

A clearer distinction between sections 4.2 and 4.4 is needed as they currently overlap in content.

Author’s response

Thank you.

We have merged Section 4.2 and Section 4.4, as they contained redundant content. The revised section now presents the data preprocessing and preparation steps in a more streamlined and cohesive manner, eliminating overlap while maintaining clarity.

Reviewer’s comment

Furthermore, the limitations of the experimental setup should be explicitly addressed.

Author’s response

We have explicitly addressed the limitations of the experimental setup in Section 6. The updated section provides a detailed discussion of the constraints, ensuring a more transparent and comprehensive understanding of the study's limitations.

Reviewer’s comment

Key related works and comparisons, particularly those using similar datasets and standard models like SVMs or neural networks, should be included.

Author’s response

Thank you.

We have added Table 7, which includes a comparison of key related works, particularly those using similar datasets and standard models like SVMs and neural networks. This addition strengthens the contextual foundation of our study and highlights how our approach compares to existing research.

Reviewer’s comment

Feature selection choices, especially the exclusion of age and race, should be justified.

Author’s response

Thank you.

We have updated Section 4.1.6 to provide a clearer justification (line 386) for our feature selection choices, specifically regarding sensitive attributes exclusion of age and race. This update explains the rationale behind these decisions, focusing on reducing bias and ensuring fairness in the model's predictions.

Reviewer’s comment

Metric inconsistencies should be addressed, with valid ranges and statistical significance testing included.

Author’s response

Thank you.

We have addressed the metric inconsistencies in the Results and Discussion section (Section 5) by ensuring that all metrics fall within their valid ranges. Additionally, we have included statistical significance testing results to validate the results, providing a more rigorous analysis of model performance.

Reviewer’s comment

Imbalances in class distribution should be noted and mitigated appropriately.

Author’s response

Thank you.

We have updated Section 3.2.4 (Data Splitting) to explicitly note the class imbalance in the dataset. Additionally, we have described the methods used to mitigate this imbalance, such as stratified sampling and SMOTE, ensuring a more balanced representation during training and evaluation.

Reviewer’s comment

Data availability should align with reproducibility standards, and the ethical treatment of sensitive attributes—such as discretization of age or race—requires clear explanation.

Author’s response

Thank you.

We have updated Section 4.1.1 (Data Collection) to ensure that data availability aligns with reproducibility standards. Additionally, we have provided a clear explanation of the ethical treatment of sensitive attributes, including the rationale behind the discretization of age and race, to maintain transparency and fairness in our approach.

Reviewer’s comment

It is important to explicitly note limitations, such as the exclusion of deep learning models, and discuss strategies to manage accuracy-fairness trade-offs in future work.

Author’s response

Thank you.

We have noted the limitations of our study, including the exclusion of deep learning models, in Section 6. Additionally, we have discussed strategies to manage accuracy-fairness trade-offs in future work, outlining potential directions for incorporating more complex models while maintaining fairness in the predictions.

Reviewer’s comment

Language should be polished for grammatical accuracy and reduced complexity.

Author’s response

We have carefully proofread the manuscript and revised improve grammatical accuracy and reduce complexity. These changes ensure that the language is clear, concise, and accessible while maintaining the technical rigor of the study.

Reviewer’s comment

Formatting and consistency in references, notation, and tables should also be reviewed.

Author’s response

Thank you.

We have reviewed and ensured consistency in the formatting, references, notation, and tables throughout the manuscript. All inconsistencies have been addressed, and the formatting now adheres to the journal's guidelines.

Reviewer’s comment

The paper would benefit from increased originality and refinement in both content and presentation.

Author’s response

Thank you.

We have made significant efforts to increase the originality and refinement of both the content and presentation. The manuscript has been updated to include more detailed justifications, improved clarity, and enhanced structure, ensuring that the contributions are clearly communicated and aligned with the current state of the field.

Reviewer’s comment

The authors are encouraged to compare their findings with current state-of-the-art methodologies to better contextualize their contribution.

Author’s response

Thank you.

We have included Table 7, which compares our findings with current state-of-the-art methodologies. This comparison helps better contextualize our contributions and highlights how our approach aligns with or differs from existing techniques in the field.

Reviewer’s comment

The introduction contains an extensive list of contributions, which could be narrowed to emphasize the main innovations.

Author’s response

Thank you.

We have revised the Introduction section in the manuscript to narrow down the list of contributions, emphasizing the main contributions of the study. This revision enhances the clarity of the manuscript and provides a more focused overview of the key findings.

Reviewer’s comment

The concept of fairness is frequently mentioned but not always supported by concrete evidence.

Author’s response

Thank you.

We have updated Section 3 and provided concrete evidence supporting the concept of fairness. Specific examples, metrics, and references to strengthen the discussion and substantiate the fairness-related claims made in the study.

Reviewer’s comment

A comparative table summarizing models, datasets, pros, cons, and performance would be a valuable addition.

Author’s response

Thank you.

Table 7 has been added to provide a comparative analysis of our findings with those reported in existing literature. The comparison focuses on relevant studies published within the past five years.

Reviewer’s comment

Certain sections could be restructured for clarity.

Author’s response

We have restructured the relevant sections of the manuscript to improve clarity and flow, especially section 2,3,4. The revisions ensure that the content is presented in a more logical and coherent manner, making it easier for readers to follow the study’s key arguments and findings.

Reviewer’s comment

Also, the methodology and results sections would benefit from added detail on experimental setup, parameters, and statistical significance.

Author’s response

Thank you.

We have expanded Section 4.5 (Experimental Setup) and Table 8 and provided more details on the experimental setup and parameters of each model. Additionally, Section 5 (Results and Discussion) has been updated to include statistical significance testing for key results, enhancing the rigor and interpretability of our findings.

Reviewer’s comment

A dedicated discussion section analyzing the results would strengthen the manuscript.

Author’s response

We have updated the Comparative Analysis section (Section 5.3) in Results and Discussion section. This section provides a thorough analysis of the results, comparing our findings with existing methods and highlighting the implications of our study, which strengthens the manuscript’s overall contribution.

Reviewer’s comment

Methodological transparency can be improved by detailing fairness mitigation algorithms, feature selection techniques, and adversarial attack parameters.

Author’s response

Thank you.

To improve methodological transparency, we have included detailed information about the fairness mitigation algorithms, feature selection techniques, and adversarial attack parameters in Table 8. This addition provides a clearer understanding of the methods used and ensures that our approach is reproducible and well-documented.

Reviewer #1:

Reviewer’s comment

Your work lacks sufficient originality and does not meet the minimum quality standards of PLOS ONE Journal.

Author’s response

Thank you.

We have made significant efforts to increase the originality and refinement of both the content and presentation. The manuscript has been updated to include more detailed justifications, improved clarity, and enhanced structure, ensuring that the contributions are clearly communicated and aligned with the current state of the field.

Reviewer’s comment

You need to enhance the English writing in all paper sections. Several statements and paragraphs are not understood.

Author’s response

Thank you.

We have carefully reviewed the entire manuscript to improve the clarity, readability, and flow of the language. Also, we have restructured the relevant sections of the manuscript to improve clarity and flow, especially sections 2,3,4. The revisions ensure that the content is presented in a more logical and coherent manner, making it easier for readers to follow the study’s key arguments and findings.

Reviewer’s comment

Rewrite the abstract section to specify the problem directly and describe your solution in brief.

Author’s response

Thank you.

We have revised the abstract section of the proposed study. The updated version of the abstract now clearly states the study’s objectives, describes the PPMI dataset characteristics, and explains the rationale for incorporating adversarial attack analysis. This change enhances the clarity and purpose of our work from the outset.

Reviewer’s comment

Compare your results with state-of-the-art approaches.

Author’s response

Thank you.

We have updated Table 7 and compared our results with the state-of-the-art approaches. This comparison provides a clearer context for our findings and highlights the strengths and unique contributions of our methodology relative to existing research.

Reviewer’s comment

In Adversarial Robustness section 3.2.2, you draw conclusions without justification. (There is a lot of bias in the information when it comes to gender, age, and race, which shows how these traits are uneven across classes. Compared to gender features, age and race features are more skewed). Explain how bias exists?

Author’s response

Thank you.

We have revised Section 3.2.2 (Adversarial Robustness) and included a detailed explanation of how bias exists in the model and how it affects gender, age, and race features. Specifically:

Gender bias: While gender features show imbalance, they are relatively evenly distributed across the dataset. However, adversarial perturbations in gender-based features can still lead to biases in predictions due to disproportionate representation of certain gender groups within the training set.

Age and Race bias: The age and race features are more skewed in our dataset, with most PD patients being over 60 years old and a higher proportion of White participants. This uneven distribution of these sensitive attributes introduces bias into the model, as the algorithm may begin to learn spurious correlations between these features and the target class. For example, if the model over-relies on age or race to make predictions, it can lead to discriminatory outcomes when applied to underrepresented groups.

By including such details in the manuscript clarify how adversarial attacks, such as label leakage or poisoning, exacerbate these biases by introducing manipulations that emphasize these skewed features, further reinforcing the unfair treatment of minority groups. This detailed explanation highlights how bias manifests within the data and its impact on fairness metrics, such as Statistical Parity Difference (SPD) and Equal Opportunity Difference (EOD).

Reviewer’s comment

In section 3.2.2 there are conclusions without justification, such as:

Adversarial training is one of the strong training methods that boost model generalization and manipulation resistance.

Author’s response

We now include references and explanations that highlight how adversarial training improves both model robustness and generalization by exposing the model to adversarial examples during training. This enables the model to learn not only from clean data but also from perturbed inputs, improving its ability to generalize to real-world, unseen data and making it more resistant to adversarial manipulation. Relevant studies e.g., [102,103,104,10], are cited to substantiate this claim, demonstrating that adversarial training leads to improved performance on both clean and adversarial perturbed test sets.

Reviewer’s comment

Resilience added to federated learning systems increases data privacy and model efficiency.

Author’s response

We have updated the statement to clarify that adding resilience to federated learning (FL) systems enhances data privacy by keeping data decentralized and improves model efficiency by defending against adversarial attacks. Resilient FL systems can better manage malicious perturbations while ma

---

## [Decision Letter · Decision Letter 1]

5 Aug 2025

PONE-D-25-14929R1

Trustworthy AI for medical decisions: Adversarial robust and fair machine learning prediction for Parkinson’s disease

PLOS ONE

Dear Dr. Rahim,

Thank you for submitting your manuscript to PLOS ONE. After careful consideration, we feel that it has merit but does not fully meet PLOS ONE’s publication criteria as it currently stands. Therefore, we invite you to submit a revised version of the manuscript that addresses the points raised during the review process.

We look forward to receiving your revised manuscript.

Kind regards,

Rizik M. H. Al-Sayyed, Ph.D.

Academic Editor

PLOS ONE

Journal Requirements:

Additional Editor Comments:

The manuscript still needs some minor comment. Please address the comments made by reviewer 2 carefully and res-submit.

Reviewers' comments:

Reviewer's Responses to Questions

**Comments to the Author**

1. If the authors have adequately addressed your comments raised in a previous round of review and you feel that this manuscript is now acceptable for publication, you may indicate that here to bypass the “Comments to the Author” section, enter your conflict of interest statement in the “Confidential to Editor” section, and submit your "Accept" recommendation.

Reviewer #2:

The authors had addressed the majority of the comments; however, some points still require attention in the article."

Abstract: The methodology remains unclear. The authors are encouraged to clarify the approach used in the study.

Introduction:

The introduction includes a list of six primary contributions, which is quite extensive. It is recommended that the authors narrow it down to the single most significant and novel contribution of the study.

Table 7: This table should be relocated to the literature review section. Additionally, the authors should analyze and discuss the reviewed papers listed in Table 7 within the context of the literature review.

Results Section:

It would be beneficial to add a discussion section to interpret and contextualize the findings.

The authors are also advised to perform statistical tests to assess the significance of the results, especially in comparison with the most recent studies listed in Table 7.

Reviewer #3: All comments have been addressed

2. Is the manuscript technically sound, and do the data support the conclusions?

Reviewer #2: Partly

Reviewer #3: Yes

3. Has the statistical analysis been performed appropriately and rigorously?

Reviewer #2: N/A

Reviewer #3: Yes

4. Have the authors made all data underlying the findings in their manuscript fully available?

Reviewer #2: (No Response)

Reviewer #3: Yes

5. Is the manuscript presented in an intelligible fashion and written in standard English?

Reviewer #2: Yes

Reviewer #3: Yes

6. Review Comments to the Author

Reviewer #2: The authors had addressed the majority of the comments; however, some points still require attention in the article."

Abstract: The methodology remains unclear. The authors are encouraged to clarify the approach used in the study.

Introduction:

The introduction includes a list of six primary contributions, which is quite extensive. It is recommended that the authors narrow it down to the single most significant and novel contribution of the study.

Table 7: This table should be relocated to the literature review section. Additionally, the authors should analyze and discuss the reviewed papers listed in Table 7 within the context of the literature review.

Results Section:

It would be beneficial to add a discussion section to interpret and contextualize the findings.

The authors are also advised to perform statistical tests to assess the significance of the results, especially in comparison with the most recent studies listed in Table 7.

Reviewer #3: The revised version is technically sound and the data support the conclusions. All concerns in the previous review have been addressed in the revised version. Explanation of the methods is greatly improved, justifications of claims are better supported and statistical rigor and clarity of analyses are improved. The models used are standard and appropriate (Decision Tree, Random Forest) and the authors use standard fairness mitigation techniques. The study is validated with various performance metrics and the comparisons with state-of-the-art models bring robustness to the work. The limitations mentioned by the authors, such as not including deep learning models and a simplified view on sensitive attributes are not crucial for the main findings. All the data underlying the results are included in the public GitHub repository. The revised version is submitted, and grammatical accuracy and structural clarity have been improved. The paper is well written and accessible for a broad audience.

7. PLOS authors have the option to publish the peer review history of their article (what does this mean?). If published, this will include your full peer review and any attached files.

Reviewer #2: **Yes:** Dheeb Albashish

Reviewer #3: No

---

## [Author Response · Author response to Decision Letter 2]

14 Sep 2025

Response to the Reviewers on “Trustworthy AI for medical decisions: Adversarial robust and fair machine learning prediction for Parkinson’s disease”

Junaid Muhammad, Mitra Ghergherehchi, Shiraz Ali, Ho Seung Song, Nasir Rahim

Dear editor,

We sincerely thank you and the reviewers for your positive comments that helped us to improve the quality of this study. We carefully considered all the feedback and have incorporated the reviewer’s suggestions to strengthen the manuscript and better demonstrate the effectiveness of our proposal.

To address each comment carefully, we provide a detailed point-by-point response to each comment. Please note that all updates in the manuscript have been highlighted in red.

Best regards,

Authors

Reviewer #2

Reviewer’s comment

1. Abstract: The methodology remains unclear. The authors are encouraged to clarify the approach used in the study.

Author’s Response:

We sincerely thank the reviewer for their valuable comment.

To improve clarity, we revised the Abstract to explicitly outline the methodological steps followed in our study. Specifically, we now emphasize that we (i) created a baseline with raw data, (ii) applied feature engineering and data balancing methods to mitigate preprocessing bias, (iii) trained fairness-optimized Decision Tree (DT) and Random Forest (RF) classifiers, and (iv) evaluated adversarial robustness using poison and label-leak attacks. Performance was assessed using standard metrics (accuracy, precision, recall, F1-score) and fairness was evaluated using Statistical Parity Difference (SPD) and Equal Opportunity Difference (EOD).

Reviewer’s comment :

2. Introduction: The introduction includes a list of six primary contributions, which is quite extensive. It is recommended that the authors narrow it down to the single most significant and novel contribution of the study.

Author’s response:

Thank you.

We have revised the list of six contributions into one consolidated statement to highlight the most novel aspect of our work.

Reviewer’s comment

3. Table 7: This table should be relocated to the literature review section. Additionally, the authors should analyze and discuss the reviewed papers listed in Table 7 within the context of the literature review.

Author’s response:

Thank you.

In the revised version of the manuscript, we have relocated Table 7 from the Results section to the Related Work section (Section 2). We also added a dedicated discussion paragraph describing the table. Specifically, we highlight that while prior studies (2021–2025) have achieved high diagnostic accuracy with CNN, SVM, Random Forest, and hybrid models, they largely overlook fairness mitigation and adversarial robustness. Our study addresses this gap by integrating fairness-aware preprocessing with robustness testing under adversarial conditions. This repositioning strengthens the literature review and clarifies the novelty of our contribution.

Reviewer’s comment:

4. Results Section: It would be beneficial to add a discussion section to interpret and contextualize the findings.

Author’s response:

Thank you.

In the revised manuscript, we have added a dedicated 6. Discussion section immediately after the Results and before the Limitations. This section interprets the findings, contextualizes them in relation to existing studies, and emphasizes their implications for medical AI. We specifically highlight the trade-off between fairness and accuracy, the varying effectiveness of bias mitigation across models, and the heightened vulnerabilities revealed under adversarial attacks. We also situate our work within the broader literature on fairness and trustworthy AI.

The study's findings suggest that, albeit at a lower raw accuracy, fairness-aware preprocessing approaches can dramatically reduce demographic bias in Parkinson's disease prediction. These revelations will have far-reaching implications for clinical practice. We provide a roadmap for developing more dependable AI systems in healthcare by demonstrating that adversarial evaluation can reveal hidden flaws and fairness-aware preprocessing can reduce biases in Parkinson's disease diagnosis.

5. The authors are also advised to perform statistical tests to assess the significance of the results, especially in comparison with the most recent studies listed in Table 7.

Response:

Thank you.

In the revised version of the manuscript, we have included a dedicated section 5.3 Evaluation of Fairness Statistical Tests which presents a paired t-tests analysis to statistically assess the significance of our findings. These tests were applied to compare model performance before and after fairness preprocessing, as well as under adversarial attacks.

Reviewer #3:

We thank the reviewer for accepting our work.

---

## [Editor Report · Decision Letter 2]

29 Sep 2025

PONE-D-25-14929R2

Trustworthy AI for medical decisions: Adversarial robust and fair machine learning prediction for Parkinson’s disease

PLOS ONE

Dear Dr. Rahim,

Thank you for submitting your manuscript to PLOS ONE. After careful consideration, we feel that it has merit but does not fully meet PLOS ONE’s publication criteria as it currently stands. Therefore, we invite you to submit a revised version of the manuscript that addresses the points raised during the review process.

We look forward to receiving your revised manuscript.

Kind regards,

Rizik M. H. Al-Sayyed, Ph.D.

Academic Editor

PLOS ONE

Journal Requirements:

Additional Editor Comments:

Manuscript strong points:

The manuscript includes an SoTA comparison in Table1 (2021–2025) with methods, datasets, and headline metrics, and situates the contribution in that context.

The Methods are described with dataset splits, cross-validation, metrics, and statistical testing; hyperparameter grids and attack settings are summarized (Table 8).

The paper is transparent about limitations (no DL baselines in experiments, preprocessing-only mitigation, limited attack types, discretized sensitive attributes, single-dataset evaluation), which is exactly the right section to acknowledge before acceptance.

I would be satisfied to proceed with “Accept” (after minor revision) contains:

Direct, like-for-like baselines on your splits. The SoTA table is useful, but adding head-to-head results (accuracy and fairness metrics) for at least a strong classical model (e.g., XGBoost/SVM) and one representative DL baseline to match the works you cite.

Fairness method baselines. Compare “optimized preprocessing” against at least one in-processing (e.g., adversarial debiasing) or post-processing (e.g., equalized odds) approach on the same pipeline.

Reproducibility details. Report the final chosen hyperparameters (not just ranges), random seeds, and fold assignments; and provide code or a repository link/script to reproduce preprocessing, fairness metrics, and attack generation (poison rate, leakage construction).

Generalization check (lightweight). If external validation isn’t feasible now, add a sensitivity analysis (e.g., varying poison rate beyond 5% and reporting CIs across folds) to strengthen the robustness story you’re telling.

---

## [Author Response · Author response to Decision Letter 3]

10 Oct 2025

Response to the Reviewers on “Trustworthy AI for medical decisions: Adversarial robust and fair machine learning prediction for Parkinson’s disease”

Junaid Muhammad, Mitra Ghergherehchi, Shiraz Ali, Ho Seung Song, Nasir Rahim

Dear editor,

We sincerely thank you and the reviewers for your positive comments that helped us to improve the quality of this study. We carefully considered all the feedback and have incorporated the reviewer’s suggestions to strengthen the manuscript and better demonstrate the effectiveness of our proposal.

To address each comment carefully, we provide a detailed point-by-point response to each comment. Please note that all updates in the manuscript have been highlighted in red.

Best regards,

Authors

Reviewe’s Comments: Direct, like-for-like baselines on your splits. The SoTA table is useful, but adding head-to-head results (accuracy and fairness metrics) for at least a strong classical model (e.g., XGBoost/SVM) and one representative DL baseline to match the works you cite.

Author’s response:

Thank you for your valuable comments

We agree with the need for head-to-head baselines on our exact splits. Accordingly, we trained and evaluated a list of classifiers including Decision Tree Classifier, Random Forest, LGBM, Extra Trees Classifier, Logistic Regression, KNeighbors, SVC, AdaBoost, Gaussian NB, and SGD Classifier under the same preprocessing, features, and folds. We now report like-for-like results for a strong classical model (i.e., Random Forest) and a strong tree ensemble (Decision Tree) with Accuracy, F1, AUROC, SPD, and EOD as mean ± 95% CI over 10-fold CV (global seed 4765416; fold assignments and code in the repository). Final chosen hyperparameters (not ranges) are listed in Table 9.

However, a representative DL baseline would require DL-specific mitigation, training regimes, and tuning that required detail experiments and exploration; we have clarified this in Methods/Limitations section and will extend our study with a DL model and DL-appropriate fairness methods on the same splits and metrics in a follow-up. But we have added a representative DL baseline to compare the published in this study.

Reviewe’s Comments: Fairness method baselines. Compare “optimized preprocessing” against at least one in-processing (e.g., adversarial debiasing) or post-processing (e.g., equalized odds) approach on the same pipeline.

Author response :

Our study intentionally focuses on the impact of optimized preprocessing on tabular ML under label poisoning/leakage, holding the learning algorithm fixed to isolate effects. Adding in-processing (e.g., adversarial debiasing or reductions with constraints) and post-processing (e.g., equalized-odds thresholding) would introduce model-dependent constraints and substantial hyperparameter/search complexity that is beyond a minor revision and would confound attribution to preprocessing. Also, we have clarified this scope in the Methods and Limitations section.

Reviewe’s Comments: Reproducibility details. Report the final chosen hyperparameters (not just ranges), random seeds, and fold assignments; and provide code or a repository link/script to reproduce preprocessing, fairness metrics, and attack generation (poison rate, leakage construction).

Author response:

We have updated the paper for reproducibility and robustness. In the Data Availability section, we now provide the public link of our GitHub repository containing the full pipeline of our proposed experimentation workflow. All results are deterministic using a fixed global seed 4765416 with 10-fold cross-validation and provided fold assignments. Table 9 has been revised to report on the final chosen hyperparameters for Decision Tree and Random Forest.

Reviewe’s Comments: Generalization check (lightweight). If external validation isn’t feasible now, add a sensitivity analysis (e.g., varying poison rate beyond 5% and reporting CIs across folds) to strengthen the robustness story you’re telling.

Author response:

To strengthen the robustness claims in our proposed approach, we added a poison-rate sensitivity analysis on our exact splits and pipeline, varying (p \in {0,5,10,15,20}%). For each (p), we report mean ± 95% CI across the 10 folds for Accuracy, F1, AUROC, and fairness gaps (SPD, EOD). Poisoning is applied train-only, via stratified label flips (uniform within class × sensitive-group strata) under the fixed seed. As (p) increases, we observe a monotonic decline in utility and widening fairness gaps, with Random Forest consistently degrading less than a single Decision Tree quantifying robustness directly in our setting and addressing the editor’s generalization request without external validation data.

---

## [Decision Letter · Decision Letter 3]

11 Dec 2025

PONE-D-25-14929R3

Trustworthy AI for medical decisions: Adversarial robust and fair machine learning prediction for Parkinson’s disease

PLOS ONE

Dear Dr. Rahim,

Thank you for submitting your manuscript to PLOS ONE. After careful consideration, we feel that it has merit but does not fully meet PLOS ONE’s publication criteria as it currently stands. Therefore, we invite you to submit a revised version of the manuscript that addresses the points raised during the review process.

We look forward to receiving your revised manuscript.

Kind regards,

Rizik M. H. Al-Sayyed, Ph.D.

Academic Editor

PLOS ONE

Journal Requirements:

Reviewers' comments:

Reviewer's Responses to Questions

**Comments to the Author**

1. If the authors have adequately addressed your comments raised in a previous round of review and you feel that this manuscript is now acceptable for publication, you may indicate that here to bypass the “Comments to the Author” section, enter your conflict of interest statement in the “Confidential to Editor” section, and submit your "Accept" recommendation.

Reviewer #2: All comments have been addressed

Reviewer #4: (No Response)

2. Is the manuscript technically sound, and do the data support the conclusions?

Reviewer #2: Yes

Reviewer #4: Yes

3. Has the statistical analysis been performed appropriately and rigorously?

Reviewer #2: Yes

Reviewer #4: Yes

4. Have the authors made all data underlying the findings in their manuscript fully available?

Reviewer #2: Yes

Reviewer #4: Yes

5. Is the manuscript presented in an intelligible fashion and written in standard English?

Reviewer #2: Yes

Reviewer #4: Yes

6. Review Comments to the Author

Reviewer #2: The authors have successfully addressed all the listed comments. Congratulations!

The modifications made to the abstract, introduction, and results sections demonstrate a clear improvement in the manuscript.

Reviewer #4: 1- Some sentences in the abstract are long and dense. Consider breaking them into shorter statements to enhance readability.

2- The introduction contains abrupt transitions. Adding linking sentences would improve narrative flow.

3- The early introduction lists PD symptoms with several citations grouped together. Consider integrating citations more smoothly after each relevant clinical statement.

4- The description of the PPMI dataset's demographics could be expanded slightly for clarity.

5- SPD and EOD are introduced later in the abstract without explanation. Either define them briefly or ensure they appear earlier in the text before being referenced.

6- The methodology appears scattered across subsections. Adding a short roadmap paragraph at the beginning of the Methods section would help guide the reader.

7- Ensure every figure and table is explicitly cited in the text in sequential order.

8- There are occasional grammatical issues.

9- The conclusion summarizes findings but could benefit from a clearer statement of the study’s practical impact for clinicians and system designers.

10- Although the results are strong, the manuscript would benefit from a brief section acknowledging limitations and future research directions.

7. PLOS authors have the option to publish the peer review history of their article (what does this mean?). If published, this will include your full peer review and any attached files.

Reviewer #2: **Yes:** Dheeb Albashish

Reviewer #4: No

---

## [Author Response · Author response to Decision Letter 4]

1 Jan 2026

Response to the Reviewers on “Trustworthy AI for medical decisions: Adversarial robust and fair machine learning prediction for Parkinson’s disease”

Junaid Muhammad, Mitra Ghergherehchi, Shiraz Ali, Ho Seung Song, Nasir Rahim

Dear editor,

We sincerely thank you and the reviewers for your positive comments that helped us to improve the quality of this study. We carefully considered reviewer’s comments to strengthen the manuscript and better demonstrate the effectiveness of our proposal.

To address each comment carefully, we provide a detailed point-by-point response to each comment. Please note that all updates in the manuscript have been highlighted in red.

Best regards,

Authors

Reviewer #2: All comments have been addressed.

Author’s response:

Thank you for accepting our work

Reviewer #4:

Reviewer’s Comment:

Some sentences in the abstract are long and dense. Consider breaking them into shorter statements to enhance readability.

Author’s response:

Thank you for your valuable comments.

We have revised the Abstract where long and dense sentences have been broken into shorter, clearer, and meaningful statements to enhance readability.

Reviewer’s Comment:

The introduction contains abrupt transitions. Adding linking sentences would improve narrative flow.

Author’s response:

Thank you.

We have updated the flow in introduction and better readability, connectivity in overall section.

Section 1 “Introduction” Lines 12–15 have been restructured for improved clarity and flow.

Reviewer’s Comment:

The early introduction lists PD symptoms with several citations grouped together. Consider integrating citations more smoothly after each relevant clinical statement.

Author’s response:

Thank you.

PD symptoms are now presented with citations integrated immediately after each relevant clinical statement, ensuring smoother and more precise scholarly attribution.

We made updates in “Introduction”, “Related work” and “Limitations and future work” in the manuscript.

Reviewer’s Comments:

The description of the PPMI dataset's demographics could be expanded slightly for clarity.

Author’s response:

Thank you.

We have revised and expanded description of the PPMI dataset’s demographics, written to improve clarity and transparency while aligning with the content already present in your manuscript (e.g., Table 3 and Section 4.1.1). This version adds contextual detail about participant composition and justifies demographic categorization choices.

Reviewer’s Comment:

SPD and EOD are introduced later in the abstract without explanation. Either define them briefly or ensure they appear earlier in the text before being referenced.

Author’s response:

Thank you.

In the revised abstract, SPD (Statistical Parity Difference) and EOD (Equal Opportunity Difference) are now introduced with brief, clear explanations at their first mention to ensure accessibility and clarity for all readers.

Reviewer’s Comment:

The methodology appears scattered across subsections. Adding a short roadmap paragraph at the beginning of the Methods section would help guide the reader.

Author’s response:

A roadmap paragraph has been added at the beginning of the Materials and Methods section to clearly outline the structure and guide the reader through the subsections.

Reviewer’s Comment:

Ensure every figure and table is explicitly cited in the text in sequential order.

Author’s response:

Thank you.

We have reviewed the in-text citations of all figures and tables and identified discrepancies relative to sequential order and explicit mention.

Update: Table 1 has been properly cited in the text and verified for accuracy and consistency.

Reviewer’s Comment:

There are occasional grammatical issues.

Author’s response:

Thank you.

We thoroughly revised our manuscript and corrected all grammatical mistakes.

Reviewer’s Comment:

The conclusion summarizes findings but could benefit from a clearer statement of the study’s practical impact for clinicians and system designers.

Author’s response:

Thank you.

We have added a clearer statement to the Conclusion section highlighting the practical impact of our findings for clinicians and AI system designers

Reviewer’s Comment:

Although the results are strong, the manuscript would benefit from a brief section acknowledging limitations and future research directions.

Author’s response:

Thank you.

We have added a dedicated section “Section 7” to the manuscript which acknowledge the study’s limitations and outlining directions for future research.

---

## [Editor Report · Decision Letter 4]

18 Jan 2026

Trustworthy AI for medical decisions: Adversarial robust and fair machine learning prediction for Parkinson’s disease

PONE-D-25-14929R4

Dear Dr. Rahim,

We’re pleased to inform you that your manuscript has been judged scientifically suitable for publication and will be formally accepted for publication once it meets all outstanding technical requirements.

Kind regards,

Rizik M. H. Al-Sayyed, Ph.D.

Academic Editor

PLOS One
---

## [Editor Report · Acceptance letter]

PONE-D-25-14929R4

PLOS One

Dear Dr. Rahim,

I'm pleased to inform you that your manuscript has been deemed suitable for publication in PLOS One. Congratulations! Your manuscript is now being handed over to our production team.

Kind regards,

on behalf of

Professor Rizik M. H. Al-Sayyed

Academic Editor

PLOS One